# Q-SAM: Unlocking Sharpness-Aware Minimization for Generalization in Offline Reinforcement Learning

**Da Wang**[1]  **Yi Ma**[1]  **Ting Guo**[2]  **Lin Li**[1]  **Wei Wei**[1]  **Jiye Liang**[1]

## Abstract

Generalization remains a central challenge in offline reinforcement learning (RL), where policies are trained solely from static datasets and must perform reliably under distribution shift. While most existing offline RL methods focus on reducing training loss using standard optimizers such as Adam, the role of loss landscape geometry − particularly sharpness − has received little attention. Sharpness-Aware Minimization (SAM) has recently shown strong generalization benefits in supervised learning by favoring flatter minima. However, directly applying SAM to offline RL is non-trivial: unlike supervised settings with ground-truth labels, offline RL relies on bootstrapped targets, making sharpness estimation noisy and often destabilizing optimization. In this paper, we revisit offline RL from an optimization perspective and investigate how sharpness-aware optimization can be made effective in this setting. We propose Q bound weighted SAM (Q-SAM), a robust and scalable framework that treats sharpness as a weighted objective and selectively prioritizes samples that are most suitable for sharpness-aware optimization based on Q bounds. By aligning the SAM objective with the characteristics of bootstrapped value estimation, Q-SAM amplifies the benefits of sharpness minimization while preserving training stability. Extensive experiments on standard offline RL benchmarks demonstrate that Q-SAM consistently improves generalization performance across diverse datasets and algorithms. Our results highlight the importance of loss sharpness in offline RL and suggest optimizer design as a promising direction for developing more robust offline RL methods.

[1]School of Computer and Information Technology, Shanxi University, Taiyuan, China [2]Data Science and Technology, North University of China, Taiyuan, China. Correspondence to: Yi Ma <mayi@sxu.edu.cn>.

*Proceedings of the 43rd International Conference on Machine Learning*, Seoul, South Korea. PMLR 306, 2026. Copyright 2026 by the author(s).

## 1. Introduction

Offline reinforcement learning (RL) (Levine et al., 2020) focuses on learning from pre-collected static datasets without requiring direct interaction with the environment. A key challenge is to address the distribution shift, particularly with the overestimation of out-of-distribution (OOD) (Fujimoto et al., 2019) data exacerbated during the extrapolation process (Kumar et al., 2019). Previous work employs techniques such as policy constraints (Fujimoto & Gu, 2021; Fakoor et al., 2021; Wu et al., 2022; Li et al., 2023b; Ran et al., 2023), value function regularization (Kumar et al., 2020b; Kostrikov et al., 2021; Ma et al., 2021), uncertainty estimation (An et al., 2021; Bai et al., 2022; Wu et al., 2021; Wang et al., 2025b; Qiao et al., 2025), and density ratio estimation (Lee et al., 2022; Sikchi et al., 2024; Mao et al., 2024b;a), aiming to obtain optimal policies from limited and suboptimal datasets.

A noteworthy recent trend seeks to avoid overly pessimistic conservatism, striving to achieve generalization to OOD data. Existing methods for improving generalization commonly relax the constraints on the dataset (Ran et al., 2023), value penalization (Lyu et al., 2022), or state-action pair support (Wu et al., 2022; Mao et al., 2023). Other data-driven methods (Qi et al., 2022; Wang et al., 2024; 2025a) model the generalization task as a domain adaptation problem. However, those mentioned above typically do not pay much attention to the optimizer choice, often defaulting to widely validated general-purpose optimizers like Adam (Kingma & Ba, 2015), with the main focus on minimizing the loss value. Crucially, established findings (Keskar et al., 2016) indicate that the complexity and non-convex nature of training loss landscapes is commonly observed in modern deep learning models. In offline RL, where no interaction with the environment occurs, simply minimizing widely used loss functions on static datasets may lead to overfitting. The potential overfitting, in turn, is insufficient for achieving satisfactory generalization in loss landscapes with a multiplicity of local and global minima (Haarnoja et al., 2017). Extensive theoretical and empirical researches (Keskar et al., 2016; Chaudhari et al., 2017; Izmailov et al., 2018; Li et al., 2018) have demonstrated the relationship between the flatness of minima and generalization, with Sharpness-Aware

Minimization (SAM) (Foret et al., 2021) emerging as one of the most promising optimizers for improving generalization by explicitly seeking flatter minima. SAM has been widely applied in supervised deep learning fields, such as computer vision (CV) (Chen et al., 2022), natural language processing (NLP) (Bahri et al., 2022), and meta-learning (Abbas et al., 2022). In contrast to these fields, however, achievements of comparable significance in finding flatter minima to train generalizable and scalable algorithms have not yet fully found their way to offline RL. The primary challenge is lacking "true labels" in the training data (i.e., the optimal policy is inaccessible), which requires approximation in a complex bootstrapping manner.

In this paper, we revisit offline RL from the perspective of optimization geometry, investigating whether the flatness-seeking capability of SAM can translate into improved generalization. In Section 3.1, we provide a detailed comparison of the policy surface, explicitly demonstrating that SAM induces smoother decision boundaries compared to Adam. However, the direct application of SAM falls short of realizing its potential. We identify a fundamental conflict between SAM's stationary objective and the non-stationary nature of bootstrapped targets: blindly pursuing flatness on such a shifting landscape can exacerbate extrapolation errors and destabilize convergence. To address this, we delve into these optimization dilemmas (Section 3.2) and propose a robust and scalable solution. Specifically, we first adopt the weighted sharpness formulation (Yue et al., 2023) to adaptively regulate the regularization strength (Section 4.1). Then, we integrate a sample-weighting strategy that leverages Q-value bounds to prioritize data points most suitable for SAM optimization (Section 4.2). These two components consist of our proposed method, termed Q bound weighted SAM (Q-SAM). Extensive experiments on standard benchmarks and small-sample settings verify the effectiveness of using SAM in offline RL to improve generalization, with the proposed method yielding significant performance gains.

Contributions of this paper can be summarized as:

1. We empirically demonstrate that the parameter-space flatness sought by SAM translates into policy smoothness in the state-action space, serving as a critical regularizer against distribution shift.

2. We identify a fundamental conflict between sharpness minimization and the bootstrapping mechanism, revealing that blindly pursuing flatness on non-stationary targets exacerbates extrapolation errors.

3. We propose Q-SAM, a robust framework that resolves this conflict by dynamically weighing sharpness objectives based on Q-value reliability. Experiments show that Q-SAM significantly enhances generalization and stability across diverse benchmarks.

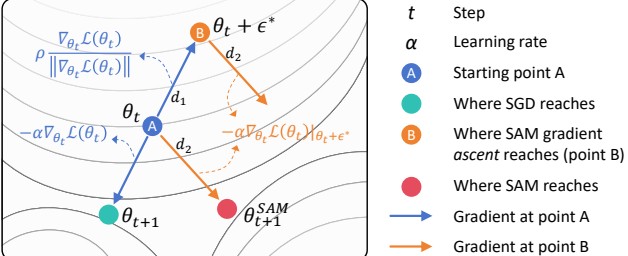

*Figure 1.* Schematic of the SAM parameter update. Standard optimizers use gradient descent to move from point A to the next parameter, $\theta_{t+1}$. SAM, however, first takes a step of size $\rho$ from point A in the direction of gradient *ascent* (denoted as $d_1$), reaching point B. At point B, SAM computes the gradient descent direction (denoted as $d_2$). Finally, starting from point A, SAM takes a step of size $\alpha$ in the $d_2$ direction and reaches the point with $\theta_{t+1}^{SAM}$.

## 2. Preliminaries

### 2.1. Offline Reinforcement Learning

Offline RL, also referred to as batch RL, aims to learn an optimal policy solely from a fixed dataset $\mathcal{D} = \{(s, a, r, s')\}$, collected by one or more behavior policies, without further interaction with the environment. The learning problem is typically formulated within the Markov Decision Process framework, defined by a tuple $(\mathcal{S}, \mathcal{A}, \mathcal{P}, r, \gamma)$, where $\mathcal{S}$ is the state space, $\mathcal{A}$ is the action space, $\mathcal{P}(s'|s, a)$ is the transition probability, $r(s, a)$ is the reward function, and $\gamma \in (0, 1)$ is the discount factor. The objective of offline RL is to learn a policy $\pi(a|s)$ that maximizes the expected cumulative reward, expressed as $J(\pi) := \frac{1}{1-\gamma} \mathbb{E}_{s \sim d^\pi, a \sim \pi(\cdot|s)} [r(s, a)]$, where $d^\pi$ denotes the discounted state visitation distribution induced by policy $\pi$. Unlike online RL (Sutton & Barto, 2018), where the agent iteratively interacts with the environment to refine its policy, offline RL faces the fundamental challenge of distributional shift: the learned policy $\pi$ may generate actions and state-action pairs that are OOD with respect to the dataset $\mathcal{D}$. This can lead to overly optimistic value function estimates, hindering policy performance.

### 2.2. Sharpness-Aware Minimization (SAM)

SAM is an advanced optimization technique designed to improve the generalization of the model by explicitly minimizing the sharpness of the loss landscape. Unlike traditional optimization algorithms such as SGD (Robbins & Monro, 1951) or Adam, which solely focus on minimizing the loss value, SAM aims to find solutions with flatter minima. The flatness of a minimum is empirically and theoretically linked to a better generalization, as flatter minima are less sensitive to perturbations in the input or model parameters.

Formally, given a loss function $\mathcal{L}(\theta)$ with model parameters $\theta$, SAM minimizes the "sharpness-aware" objective:

$$\mathcal{L}_{SAM}(\theta) = \max_{\|\epsilon\| \leq \rho} \mathcal{L}(\theta + \epsilon), \qquad (1)$$

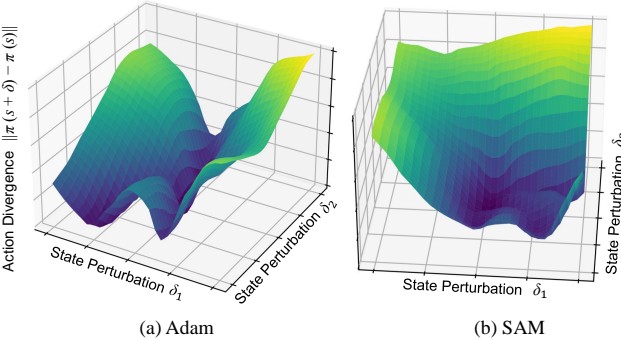

(a) Adam      (b) SAM

*Figure 2.* Visualization of the local policy landscape. The plots depict the sensitivity of the converged policy $\pi_\phi$ to state perturbations. We fix the policy parameters and introduce random perturbations to the input states. The $z$-axis represents the normalized Euclidean distance between the action generated from the perturbed state and the original optimal action (i.e., $\|\pi(s+\delta)-\pi(s)\|$). The concave minima correspond to the optimal actions for unperturbed states. Adam (a) exhibits a sharp, erratic surface, while SAM (b) induces a significantly flatter and smoother landscape.

where $\epsilon$ is a perturbation vector constrained by a norm bound $\rho$. This objective encourages the model to achieve a low loss not only in $\theta$ but also in its vicinity, effectively flattening the loss landscape. To solve the inner maximization problem efficiently, SAM approximates it using a first-order Taylor expansion around $\theta$. Specifically, for small perturbations $\epsilon$, the loss $\mathcal{L}(\theta+\epsilon)$ can be approximated as $\mathcal{L}(\theta+\epsilon) \approx \mathcal{L}(\theta) + \epsilon^\top \nabla_\theta \mathcal{L}(\theta)$. By maximizing this approximation under the norm constraint $\|\epsilon\| \leq \rho$, the optimal perturbation $\epsilon^*$ is:

$$\epsilon^* \approx \arg \max_{\|\epsilon\| \leq \rho} \mathcal{L}(\theta) + \epsilon^\top \nabla_\theta \mathcal{L}(\theta) = \rho \frac{\nabla_\theta \mathcal{L}(\theta)}{\|\nabla_\theta \mathcal{L}(\theta)\|}. \quad (2)$$

Substituting $\epsilon^*$ back into the SAM objective, the perturbed parameters $\theta + \epsilon^*$ are used to compute the updated gradient:

$$\nabla_\theta \mathcal{L}_{SAM}(\theta) \approx \nabla_\theta \mathcal{L}(\theta + \epsilon^*) \quad (3)$$
$$= \nabla_\theta \mathcal{L}(\theta)|_{\theta+\epsilon^*} + \frac{d\epsilon^*}{d\theta} \nabla_\theta \mathcal{L}(\theta)|_{\theta+\epsilon^*} \approx \nabla_\theta \mathcal{L}(\theta)|_{\theta+\epsilon^*},$$

where the second approximation is for accelerating the computation by dropping the second-order terms. Figure 1 illustrates the update of the SAM parameter.

# 3. SAM in Offline RL: Smoothing Effects and Optimization Dilemmas

SAM has emerged as a promising optimizer for improving generalization by explicitly seeking flatter minima. However, its application to offline RL remains underexplored and presents unique challenges compared to supervised learning. In this section, we aim to investigate the impact of optimizer selection on policy geometry and analyze the fundamental conflicts that arise when sharpness-aware updates interact with the bootstrapping mechanism.

*Table 1.* Quantitative comparison of policy smoothness. A lower **Local Action Sensitivity (LAS)** score implies a smaller local Lipschitz constant and greater robustness to state perturbations.

| Optimizer | LAS ($\downarrow$) | Geometric Interpretation |
|---|---|---|
| Adam | 1.36 | High Sensitivity (Sharp) |
| **SAM** | **0.67** | **Low Sensitivity (Smooth)** |

## 3.1. Smoothing Effects: Linking Parameter Flatness to Policy Smoothness

A core ambiguity in applying SAM to RL is the distinction between the parameter space (where optimization occurs) and the state-action space (where the policy operates). While SAM explicitly minimizes curvature with respect to parameters $\theta$, we argue that this optimization induces necessary regularization in the policy's functional behavior.

Theoretical insights suggest that flat minima in parameter space often correspond to functions with lower local complexity. In the context of offline RL, we hypothesize that this translates to policy smoothness − a critical property for mitigating distribution shift. A "sharp" policy may produce drastically different actions given slight perturbations in the state. This is particularly dangerous in offline settings, as erratic actions can query the Q-function on OOD inputs, leading to extrapolation errors.

We empirically verify this hypothesis by visualizing the policy surface (Figure 2). The experiments are conducted on D4RL (Fu et al., 2020) gym MuJoCo-v2 (hopper-medium-replay) using TD3BC (Fujimoto & Gu, 2021) as the baseline. We fix the converged policy network and observe the local action generation surface by perturbing a batch of states.

- **Observation**: As visualized in Figure 2, the policy surface obtained by Adam exhibits greater sharpness, whereas SAM induces a significantly flatter and smoother landscape.

- **Quantitative Verification**: To rigorously quantify this visual observation, we introduce the Local Action Sensitivity (LAS) metric. Drawing on the concept of local Lipschitz continuity (Asadi et al., 2018) and smoothness regularization in offline RL (Sinha et al., 2022), LAS measures the expected rate of action change under local state perturbations:

$$\mathcal{M}(\pi) = \mathbb{E}_{s \sim \mathcal{D}, \delta \sim \mathcal{N}(0, \sigma^2 I)} \left[ \frac{\|\pi(s+\delta) - \pi(s)\|_2}{\|\delta\|_2} \right]$$

The quantitative results are reported in Table 1. Adam yields a high LAS score of 1.36, reflecting high sensitivity to input noise. In contrast, SAM lowers the score to 0.67 ($\approx 50\%$ reduction), confirming that it explicitly constrains the policy's local Lipschitz constant.

Collectively, the visual landscape and the quantitative LAS metric suggest that SAM acts as a powerful smoothness regularizer. By ensuring a smoother mapping from states to actions, SAM imposes an implicit behavior constraint that prevents erratic action shifts on near-distribution states. This effectively mitigates the risk of the policy drifting into OOD regions where value estimates are unreliable, thereby shielding the agent from extrapolation errors and improving adaptability to unseen data.

### 3.2. Optimization Dilemmas: The Conflict with Bootstrapping Objectives

Despite the desirable smoothing effects, directly applying SAM to offline RL presents fundamental challenges rooted in the bootstrapping mechanism. Unlike supervised learning where labels are fixed, offline RL relies on bootstrapped targets that evolve during training, creating a non-stationary loss landscape that inherently conflicts with SAM's objective.

To formally analyze the mechanics of this conflict, we first revisit the surrogate gap that agrees with sharpness:

$$g(\theta) := \underbrace{\max_{\|\epsilon\| \leq \rho} \mathcal{L}(\theta + \epsilon)}_{\mathcal{L}_{SAM}(\theta)} - \mathcal{L}(\theta). \qquad (4)$$

Intuitively, the sharpness-aware gap $g(\theta)$ represents the difference between the maximum loss within a neighborhood and the loss at the central point. It serves as an approximation of the principal eigenvalue of the Hessian at the local minimum (Zhuang et al., 2022). Established evidence has demonstrated that $g(\theta)$ is indeed an effective metric of sharpness and that minimizing $g(\theta)$ can identify regions with flat loss surfaces (Zhuang et al., 2022). Unfortunately, $g(\theta)$ alone can only help find flatter regions rather than true minima, potentially leading to convergence at points with high loss. To address this, SAM adopts $g(\theta) + \mathcal{L}(\theta)$ as the loss function. This approach can be regarded as a trade-off between achieving a flatter surface and identifying lower minima by assigning equal weight to $g(\theta)$ and $\mathcal{L}(\theta)$.

However, when it comes to policy optimization in offline RL, the above $\mathcal{L}(\theta)$ can be replaced with the training loss of the policy network $\pi$ with parameter $\phi$, i.e., $\mathcal{L}_{SAM}(\phi) := g(\phi) + \mathcal{L}(\pi_\phi)$. It is well known that the optimal policy $\pi^*$ satisfies $\pi^*(s) = \pi_{Q^*}(s) := \arg\max_{a \in \mathcal{A}} Q^*(s, a)$, and the optimal state-action value function or $Q$ function $Q^*(s, a)$ obtained by the *Bellman optimality equation* $Q^*(s, a) = \mathcal{T}Q^*(s, a)$, where $\mathcal{T}Q^*(s, a) = \mathbb{E}_{s' \sim \mathcal{P}(\cdot|s,a)}[r(s, a) + \gamma \max_{a' \sim \pi(\cdot|s')} Q(s', a')]$ and a general form of *Bellman error* induced by the Bellman backup process can be expressed as $\mathcal{E} = \mathbb{E}\left[(Q(s, a) - (r + \gamma \max_{a'} Q(s', a')))^2\right]$. Since the optimal policy $\pi^*$ cannot be accessed in practice, we can only use the aforementioned bootstrapping method,

backtracking with predicted values over existing data, to approximate the true values. However, due to the limited nature of the offline dataset, the behavioral policy generally fails to cover all possible state-action pairs. As a result, predicting for unseen state-action pairs inevitably leads to *extrapolation errors*, which are exacerbated by the bootstrapping process, further increasing the Bellman error (Kumar et al., 2019). In addition, it is important to note that **we do not apply SAM optimization to the value function** because optimizing Q-values that are not minima can easily lead to the accumulation of Bellman errors. Exploring SAM optimization for the value function is an interesting direction for future research.

Recall that SAM minimizes the objective $\mathcal{L}_{SAM}(\phi) := g(\phi) + \mathcal{L}(\pi_\phi)$. However, given the specific issue of extrapolation error in offline RL, the interaction with bootstrapping creates a non-stationary loss landscape. This dynamic environment conflicts with SAM's objective in two specific ways:

1. **Exacerbated Extrapolation and Bellman Errors via Flatness Preference.** Applying SAM essentially seeks a trade-off between flatness ($g(\phi)$) and the loss minimum ($\mathcal{L}(\pi_\phi)$). In supervised learning, this trade-off improves generalization. However, in offline RL, prioritizing flatness prevents the policy from fully converging to the global optimum of the current value estimates. This deviation from the optimal policy $\pi^*$ is critical: it indirectly exacerbates both extrapolation errors and Bellman errors during the bootstrapping process, as the policy may drift into regions where value estimates are less accurate.

2. **Ineffective Optimization due to Asynchronous Convergence.** The second challenge arises from the accumulation of errors during bootstrapping, which causes Q-values of different samples to converge at significantly different rates. Samples with smaller Bellman errors (often near the end of trajectories) converge to reliable values faster, whereas others remain unstable for longer periods. Blindly applying SAM to all samples ignores this disparity. SAM fails to perform effectively when updates are unstable; forcing sharpness minimization on samples with large, fluctuating Bellman errors can hinder the search for true flat minima and disrupt training stability.

These challenges highlight a critical dilemma: while we desire the smoothing effects of SAM for generalization, its blind pursuit of flatness can interfere with the delicate convergence of bootstrapped value iteration. Such a conflict necessitates a tailored approach − Q-SAM − which we introduce in the following section to balance sharpness minimization with the reliability of value estimates.

# 4. Q bound weighted SAM for Offline RL

In this section, we present Q bound weighted Sharpness-Aware Minimization (Q-SAM), a robust framework tailored for offline RL. Q-SAM leverages two mechanisms — *Weighted Sharpness* and *Weighted Samples* — to address the unique optimization challenges in offline settings.

The *Weighted Sharpness* mechanism applies a scaling factor to the sharpness term, reducing its influence on the search for minima and mitigating issues such as extrapolation error and Bellman error. At the same time, the *Weighted Samples* component leverages Q bound to quantify the accumulation of Bellman error across samples. By assigning weights based on these Q bounds, it prioritizes faster-converging samples during SAM optimization, aligning the learning process with the varying dynamics of Bellman error propagation. These two components complement each other: the first assigns a weight to the sharpness term, while the second identifies which samples benefit more from controlling sharpness, ensuring a more adaptive optimization process. The following sections will detail the formulation and implementation of each component in Q-SAM.

## 4.1. SAM with Weighted Sharpness

For the first challenge, we argue that using SAM in offline RL should prioritize preserving the search for minima rather than blindly pursuing flatness. It is easy to see that SAM's trade-off process can be understood as assigning equal weights to $g(\phi)$ and $\mathcal{L}(\pi_\phi)$. Adopting the formulation from WSAM (Weighted Sharpness-Aware Minimization) (Yue et al., 2023), we incorporate the weighted sharpness term $\frac{\lambda}{1-\lambda} g(\phi)$, where the hyperparameter $\lambda \in [0,1)$ controls the weight of sharpness. Similar to $\mathcal{L}_{SAM}(\phi)$, the formal definition of $\mathcal{L}_{WSAM}(\phi)$, which composes of a vanilla loss and a sharpness term, defined as:

$$\mathcal{L}_{WSAM}(\phi) := \frac{\lambda}{1-\lambda} g(\phi) + \mathcal{L}(\pi_\phi). \quad (5)$$

The value of $\lambda$ determines the direction of the gradient computed under SAM's weight perturbation. $\lambda = 0$ and $\lambda = 0.5$ correspond to the vanilla loss ($\mathcal{L}(\pi_\phi)$) and the SAM loss ($\mathcal{L}_{SAM}(\phi)$), respectively. When $\lambda < 0.5$, $\mathcal{L}_{WSAM}(\phi)$ assigns lower weights to sharpness, guiding updates toward minima with higher curvature, making it more aggressive compared to SAM. Conversely, $\mathcal{L}_{WSAM}(\phi)$ assigns higher weights to sharpness when $\lambda > 0.5$ (see Figure 3).

Notably, since offline data is often suboptimal, over-fitting to specific minima risks trapping the policy in local optima. Consequently, the sharpness weight should be tunable based on task demands. While this work focuses on validating the effectiveness of the weighted sharpness formulation, exploring adaptive weighting mechanisms remains a promising direction for future research.

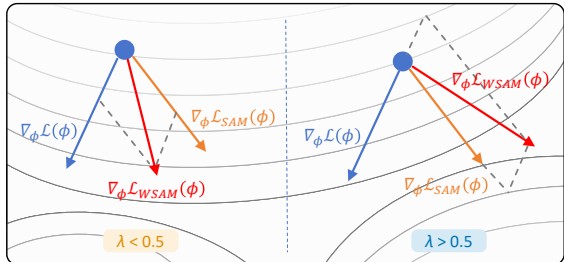

*Figure 3.* Schematic of the weighted SAM parameter update. The different values of the $\lambda$ correspond to varying preferences for sharpness, particularly influencing the direction of the gradient.

## 4.2. SAM with Weighted Samples

Section 4.1 intuitively examines the impact of directly applying SAM to offline RL. However, a deeper analysis from the perspective of offline RL reveals inherent factors that hinder the effective application of SAM. Since Bellman error tends to accumulate throughout the Bellman backup process, the convergence of earlier samples in the temporal sequence is influenced by the convergence behavior of subsequent samples. Applying SAM optimization to samples that are still undergoing unstable updates during training may not only hinder the discovery of effective flat minima but also exacerbate the accumulation of errors. Therefore, for the second challenge, we prioritize applying SAM optimization to samples that tend to quickly converge toward the minima. Intuitively, we can prioritize samples closer to optimal $Q^*$ by measuring $|Q - Q^*|$, as these samples are more likely to converge effectively. To this end, we consider a suitable method, DisCor (Kumar et al., 2020a), which computes a tractable upper bound on $|Q - Q^*|$ and uses it to reweight the transitions used for training. Specifically, the tractable upper bound on $|Q - Q^*|$, constructed only from prior $Q$ function iterates, $Q_0, \cdots, Q_{k-1}$, is defined as:

$$\Delta_k = \sum_{i=1}^{k} \gamma^{k-i} \left( \prod_{j=i}^{k-1} P^{\pi_j} \right) |Q_i - \mathcal{T}Q_{i-1}|$$

$$\Longrightarrow \Delta_k = \underbrace{|Q_k - \mathcal{T}Q_{k-1}|}_{\text{current residual}} + \underbrace{\gamma P^{\pi_{k-1}} \Delta_{k-1}}_{\text{discounted future error}}, \quad (6)$$

where $P^{\pi_{k-1}} \Delta_{k-1}$ corresponds to the estimated upper bound on the error of the target values for the current transition, due to the backup operator $P^{\pi_{k-1}}$ at iteration $k$ ($k \in \{1, \cdots, N\}$). The first term captures the immediate Bellman residual at the current state-action pair, while the second term propagates the error from subsequent states via the backup operator $P^{\pi_{k-1}}$, discounted by $\gamma$. This recursive structure aggregates uncertainty backward through trajectories without requiring knowledge of $Q^*$. Derived from the aforementioned error upper bound, the sample weighting

*Table 2.* Average normalized scores over five seeds. We **bold** the highest mean.

| Dataset | TD3BC | TD3BC + SAM | TD3BC + Q-SAM | CQL | CQL + SAM | CQL + Q-SAM | IQL | IQL + SAM | IQL + Q-SAM |
|---|---|---|---|---|---|---|---|---|---|
| halfcheetah-m | 48.1 | **48.5** | 48.4$\pm$0.4 | 47.0 | 47.1 | **47.3**$\pm$0.2 | 48.3 | 48.4 | **49.2**$\pm$0.1 |
| halfcheetah-mr | 44.8 | 44.6 | **45.3**$\pm$0.4 | 45.0 | 45.5 | **45.6**$\pm$0.2 | 44.5 | 44.6 | **45.1**$\pm$0.2 |
| halfcheetah-me | 90.8 | 90.3 | **93.9**$\pm$4.6 | 95.6 | 94.1 | **95.6**$\pm$0.5 | 94.7 | 95.0 | **95.9**$\pm$0.4 |
| hopper-m | 60.4 | 60.9 | **65.7**$\pm$2.6 | 59.1 | 59.7 | **70.3**$\pm$2.5 | 67.5 | 68.5 | **75.3**$\pm$2.2 |
| hopper-mr | 64.4 | 69.4 | **87.3**$\pm$5.0 | 95.1 | 94.9 | **95.5**$\pm$5.6 | 97.4 | 100.5 | **103.0**$\pm$5.5 |
| hopper-me | 101.2 | 101.5 | **105.1**$\pm$8.4 | 99.3 | 108.7 | **109.8**$\pm$1.7 | 107.4 | 109.4 | **113.4**$\pm$5.1 |
| walker2d-m | 82.7 | 82.9 | **85.4**$\pm$1.0 | 80.8 | 81.7 | **84.2**$\pm$1.2 | 80.9 | 86.9 | **87.6**$\pm$3.0 |
| walker2d-mr | 85.6 | 86.7 | **90.1**$\pm$1.0 | 73.1 | 74.9 | **84.4**$\pm$4.2 | 82.2 | 87.2 | **90.9**$\pm$4.7 |
| walker2d-me | 110.0 | 109.0 | **111.5**$\pm$0.2 | 109.5 | 109.6 | **112.3**$\pm$0.3 | 111.7 | 111.3 | **111.7**$\pm$1.0 |
| Total | 688.0 | 693.8 | **732.7** | 704.5 | 716.2 | **745.0** | 734.6 | 751.8 | **772.1** |
| antmaze-u | 70.8 | 80.3 | **92.8**$\pm$2.5 | 92.8 | 92.3 | **93.3**$\pm$3.3 | 77.0 | 77.5 | **81.3**$\pm$6.8 |
| antmaze-ud | 44.8 | 52.8 | **66.3**$\pm$15.5 | 37.3 | 36.3 | **38.3**$\pm$4.1 | 54.3 | 56.7 | **65.9**$\pm$6.3 |
| antmaze-mp | 0.3 | 0.3 | **3.8**$\pm$0.9 | 65.8 | 68.5 | **71.8**$\pm$7.6 | 65.8 | 68.5 | **80.8**$\pm$8.5 |
| antmaze-md | 0.3 | 0.3 | **0.5**$\pm$0.3 | 67.3 | 68.1 | **68.5**$\pm$2.5 | 73.8 | 75.9 | **81.2**$\pm$5.5 |
| antmaze-lp | 0.0 | 0.0 | **0.0**$\pm$0.0 | 20.8 | 28.0 | **33.8**$\pm$6.2 | 42.0 | 48.5 | **63.3**$\pm$7.6 |
| antmaze-ld | 0.0 | 0.0 | **0.0**$\pm$0.0 | 20.5 | 23.3 | **29.3**$\pm$6.4 | 30.3 | 30.5 | **30.8**$\pm$3.9 |
| Total | 116.2 | 133.7 | **163.4** | 304.5 | 316.5 | **335.0** | 343.2 | 357.6 | **403.3** |

criterion reads:

$$w_k(s,a) \propto \exp\left(-\frac{\gamma \left[P^{\pi_{k-1}}\Delta_{k-1}\right](s,a)}{\tau}\right), \quad (7)$$

where $\tau > 1$ is a scalar. Intuitively, the weight $w_k$ can reduce the influence of transitions where the bootstrapped target Q-value has a high estimated error relative to $Q^*$. This approach effectively directs the learning process toward samples with more accurate supervision (i.e., reliable target values), which are precisely the samples that we expect to converge faster and be better suited for SAM optimization. In addition to a standard Q-function, its needed to train another parametric model to estimate $\Delta_k(s,a)$ at each state-action pair. Section B.5 provides a detailed description of the computation of $w_k$. After obtained the weight $w_k$, one can use it to minimize the Bellman error for $Q_\theta$:

$$\theta_{k+1} \leftarrow \operatorname*{argmin}_\theta \frac{1}{N} \sum_i^N w_k(s_i, a_i)(Q_\theta(s_i, a_i) - y_i)^2, \quad (8)$$

where $y_i = r_i + \gamma \max_{a'} Q_{k-1}(s_i', a')$ is the output of the target network.

### 4.3. Connection between the Two Components

These two mechanisms constitute a "filter-then-optimize" cascading system. Specifically, *Weighted Samples* reweights the data distribution according to the cumulative Bellman error $\Delta_k$, identifying reliable samples where Q-values have converged (typically located at trajectory ends or leaf nodes). *Weighted Sharpness* then seeks flat minima in the parameter space using these reliable samples, where its primary role is to guide the direction of the gradient while its secondary

*Table 3.* Comparison with several generalization methods.

| Dataset | POR | DOGE | TSRL | IQL + Q-SAM |
|---|---|---|---|---|
| halfcheetah-m | 48.8 | 45.3 | 48.2 | **49.2 $\pm$ 0.1** |
| halfcheetah-mr | 43.5 | 42.8 | 42.2 | **45.1 $\pm$ 0.2** |
| halfcheetah-me | 94.7 | 78.7 | 92.0 | **95.9 $\pm$ 0.4** |
| hopper-m | 78.6 | **98.6** | 86.7 | 75.3 $\pm$ 2.2 |
| hopper-mr | 98.9 | 76.2 | 78.7 | **103.0 $\pm$ 5.5** |
| hopper-me | 90.0 | 102.7 | 95.9 | **113.4 $\pm$ 5.1** |
| walker2d-m | 81.1 | 86.8 | 77.5 | **87.6 $\pm$ 3.0** |
| walker2d-mr | 76.6 | 87.3 | 66.1 | **90.9 $\pm$ 4.7** |
| walker2d-me | 109.1 | 110.4 | 109.8 | **111.7 $\pm$ 1.0** |
| Total | 721.3 | 728.8 | 697.1 | **772.1** |

effect is to adjust the gradient's magnitude, thereby shaping the optimization objective's preference for flatter regions.

It is worth emphasizing that both components share the theoretical foundation of Q-value uncertainty quantification via $\Delta_k$, while operating orthogonally: one regularizes the data distribution for value learning, the other the parameter landscape for policy optimization. This orthogonality means that *Weighted Samples* can be easily incorporated into *Weighted Sharpness* to create a more robust combination. Together, these distinct weighting schemes with different objectives and forms constitute our novel method, Q-SAM. Q-SAM does not impose complex constraints or require additional parameters, allowing it to seamlessly integrate into various offline RL frameworks. This ensures its generality and scalability across different tasks and policy frameworks. Algorithm 1 presents a concise version of Q-SAM. See Section A for the algorithm details and Section B.5 for the parameter analysis.

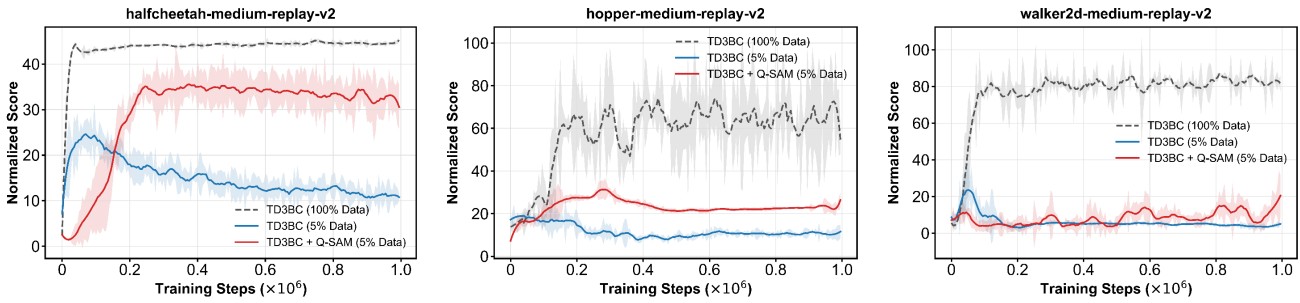

*Figure 4.* Performance comparison on full (100%) versus small-sample (5%) datasets.

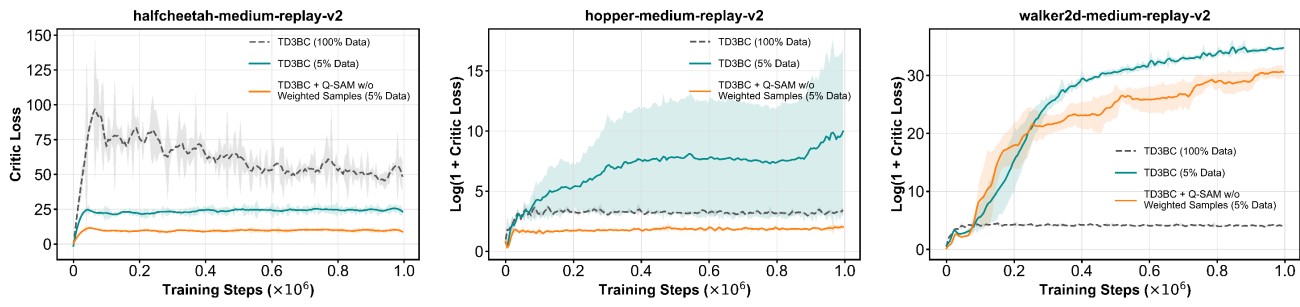

*Figure 5.* Critic loss values in the small-sample experiment.

**Algorithm 1** Q-SAM: A Concise Version

**Input:** Offline dataset $\mathcal{D}$, Q-network $Q_\theta$, error model $\Delta_\varphi$, policy $\pi_\phi$, parameters $\lambda, \rho, \alpha$.

**for** iteration $k$ in $\{1, \cdots, N\}$ **do**

Sample batch $\mathcal{B} = \{(s_i, a_i)\}_{i=1}^{b} \sim \mathcal{D}$;

# Stage 1: Weighted Samples (for Critic)

Compute $w_k(s, a)$ using Equation (7);

**Update Critic**:

$\theta_{k+1} \leftarrow \underset{\theta}{\operatorname{argmin}} \frac{1}{N} \sum_i^N w_k(s_i, a_i)(Q_\theta(s_i, a_i) - y_i)^2$;

# Stage 2: Weighted Sharpness (for Actor)

Compute the gradient $g_t^{Adam}$ and the perturbed policy gradient $g^{SAM}$ using the same weighted batch;

**Update Actor**:

$\phi_{k+1} = \phi_k - \alpha \left( \frac{\lambda}{1-\lambda} g_k^{SAM} + \frac{1-2\lambda}{1-\lambda} g_k^{Adam} \right)$;

**end for**

## 5. Experiments

In this section, we evaluate Q-SAM on performance, generalization, Bellman error reduction, ablation studies, and fairness analysis.

### 5.1. Main Results

We evaluate our idea by replacing the widely used optimizer Adam in strong baselines TD3BC (Fujimoto & Gu, 2021), CQL (Kumar et al., 2020b), and IQL (Kostrikov

et al., 2021) with SAM and Q-SAM, through a series of experiments on D4RL (Fu et al., 2020) gym MuJoCo-v2 and AntMaze-v2 datasets. We report the average normalized results of the final ten evaluations at 1M gradient step across five random seeds. Note that due to the space limitation, we abbreviate the dataset names in some tables as follows: {medium → m, medium-replay → mr, medium-expert → me} (MuJoCo-v2), and {umaze → u, umaze-diverse → ud, medium-play → mp, medium-diverse → md, large-play → lp, large-diverse → ld} (AntMaze-v2).

As shown in Table 2, the baselines exhibit varying performance gains across most tasks when applying SAM (+ SAM) and Q-SAM (+ Q-SAM), with a particularly significant improvement using the latter method. While SAM proves effective in some tasks, its overall impact is largely negligible in many cases. This suggests that the direct application of SAM may have limited potential, which likely explains its relatively low attention in previous studies. In contrast, our Q-SAM method provides a practical solution for effectively applying SAM. Beyond modest improvements in specific tasks, Q-SAM achieves remarkable performance across the majority of tasks, with breakthrough results particularly in several AntMaze tasks. Interestingly, both SAM and Q-SAM also demonstrate lower standard deviations (see complete results with standard deviations in Section B.2), which can be attributed to SAM's ability to reduce the model's sensitivity to local minima. By guiding the optimization process toward flatter loss regions, typically linked to better generalization, SAM helps the model achieve greater robustness in the testing environment.

*Table 4.* Average normalized scores over five seeds. "Drop" means the decrease in mean values.

| Dataset | CQL + Q-SAM | CQL + Q-SAM w/o Weighted Sharpness | | CQL + Q-SAM w/o Weighted Samples | |
|---|---|---|---|---|---|
| | | Score | Drop | Score | Drop |
| halfcheetah-m | 47.3±0.2 | 47.1±0.2 | 0.2 | 47.2±0.2 | 0.1 |
| halfcheetah-mr | 45.6±0.2 | 45.4±0.2 | 0.2 | 45.7±0.2 | -0.1 |
| halfcheetah-me | 95.6±0.5 | 95.6±0.4 | 0.0 | 95.9±0.4 | -0.3 |
| hopper-m | 70.3±2.5 | 67.2±2.4 | 3.1 | 65.7±1.2 | 4.6 |
| hopper-mr | 95.5±5.6 | 95.4±5.4 | 0.1 | 95.5±5.3 | 0.0 |
| hopper-me | 109.8±1.7 | 107.5±2.5 | 2.3 | 109.6±1.7 | 0.2 |
| walker2d-m | 84.2±1.2 | 81.9±1.2 | 2.3 | 83.2±0.2 | 1.0 |
| walker2d-mr | 84.4±4.2 | 80.2±5.5 | 4.2 | 82.2±4.0 | 2.2 |
| walker2d-me | 112.3±0.3 | 110.5±0.3 | 1.8 | 111.5±0.3 | 0.8 |
| Total | 745.0±16.4 | 730.8±18.1 | *14.2* | 736.5±13.5 | *8.5* |

*Table 5.* Comparison of running time and performance of different algorithms. (Time unit: minutes)

| Algorithm | Steps | halfcheetah-mr | | hopper-mr | | walker2d-mr | |
|---|---|---|---|---|---|---|---|
| | | Runtime | Score | Runtime | Score | Runtime | Score |
| TD3BC | 1M | 142.1 | 44.8±0.6 | 137.3 | 64.4±21.5 | 135.7 | 85.6±4.0 |
| | 2M | 290.7 | 45.0±0.4 | 271.5 | 65.4±21.7 | 261.4 | 68.5±11.4 |
| TD3BC + Q-SAM | 1M | 276.9 | **45.3±0.4** | 296.1 | **87.3±5.0** | 298.1 | **90.1±1.0** |

## 5.2. Validation Experiments

**Generalization**. To better verified our Q-SAM's generalization ability, inspired by (Cheng et al., 2023), we conduct a small-sample experiment by reducing the training dataset to 5% of the full data. We choose TD3BC as the baseline algorithm and the learning curves of different algorithms are shown in Figure 4. Despite a significant reduction in training data, our method still demonstrates competent performance, especially on the *halfcheetah-mr* task, which baseline algorithms fail to achieve. Fewer training data implies that the testing phase encounters more unknown scenarios, and the fact that our method still achieves effective results under challenging conditions demonstrates its superior generalization capability. We provide full results in Section B.3.

In addition, we include a comparison between our method and several representative generalization methods POR (Xu et al., 2022), DOGE (Li et al., 2023a), and TSRL (Cheng et al., 2023). As shown in Table 3, simply integrating Q-SAM into the IQL already yields competitive performance, which demonstrates the effectiveness of our approach.

**Reducing Bellman Errors**. We also provide experimental evidence in Figure 5 to support the claim that *Weighted Sharpness reduces extrapolation and Bellman errors*. Given that the Bellman error is defined as $\mathcal{E} = \mathbb{E}\left[(Q(s, a) - (r + \gamma \max_{a'} Q(s', a')))^2\right]$, the critic loss value can serve as a statistical measure of the Bellman error, where Log(1 + Critic Loss) indicates taking the logarithm

of the loss value to address scenarios of loss explosion. We remove the *Weighted Samples* from Q-SAM to better emphasize the impact of **Weighted Sharpness** in reducing Bellman error. The results show that Weighted Sharpness effectively reduces Bellman errors, and provide strong evidence that our method can achieve competitive performance.

## 5.3. Ablation Studies

A key question to address is whether the individual components of Q-SAM alone can achieve comparable performance. We conduct ablation studies to demonstrate the overall effectiveness of Q-SAM and report the performance in Table 4 ("w/o" means "without" and other results are listed in Section B.4). The ablation result shows that both components contribute to improving performance in most cases. The *Weighted Sharpness* component, which supports the SAM optimizer, has a greater impact, with its removal causing a larger performance decline compared to Q-SAM.

As for the *Weighted Samples* component, its primary role is to identify samples that should be prioritized for SAM optimization, thereby helping the SAM optimizer perform better. When the *Weighted Samples* component alone, it tends to focus more on learning the policy from the training data, with a smaller effect on improving generalization. The full Q-SAM system substantially exceeds either component individually, confirming a synergistic relationship where sample weighting stabilizes the optimization landscape for effective sharpness minimization.

## 5.4. Analysis of Unfair Comparison and Runtime

Regarding the concerns about computational overhead from SAM's nested optimization structure, we provide a fair comparison of runtime consumption and performance in Table 5. Specifically, we report the performance of TD3BC trained with double the standard budget (2M steps) and compare it against TD3BC + Q-SAM trained with the standard budget (1M steps). While SAM inherently doubles the compute per training step due to its double backward passes, Table 5 confirms that the baseline algorithm fails to achieve the performance breakthroughs that Q-SAM delivers even when granted this doubled computational budget. This finding provides strong evidence that the performance gains of Q-SAM stem fundamentally from the flat minima geometry induced by sharpness-aware optimization, rather than from merely increasing the training budget.

Indeed, the baseline tends to converge quickly, and simply doubling the training steps without altering the optimization landscape does not lead to any meaningful performance improvement. In contrast, Q-SAM achieves a substantial breakthrough in performance − at the cost of extra computation per step − that the baseline cannot match even with extended training. Given the "train-once-deploy-multiple" nature of offline RL, where the training cost is amortized across numerous deployment cycles, this additional computational overhead represents a worthwhile trade-off for the substantially improved deployment reliability and generalization conferred by flat-minima policies.

## 6. Conclusion

In this work, we revisited the optimization landscape of offline RL, revealing a **fundamental conflict between generic sharpness minimization and the non-stationary bootstrapping mechanism**. We proposed Q-SAM to harmonize these objectives. By leveraging Q-value bounds to dynamically weigh both the sharpness penalty and training samples, Q-SAM explicitly **induces policy smoothness without exacerbating extrapolation errors**. Our extensive experiments demonstrate that Q-SAM achieves superior generalization and stability across diverse benchmarks. This study highlights that **controlling optimization geometry is a vital frontier for offline RL**, offering a promising direction for designing more robust algorithms beyond standard constraints.

A limitation of this work is that Q-SAM is currently restricted to the policy network. We observed that applying SAM directly to the value function can exacerbate training instability due to the propagation of Bellman errors. Developing specialized regularization to stabilize value optimization under SAM remains a non-trivial open challenge. Furthermore, while the current $\lambda$-based weighting effectively balances flatness and convergence, offline-RL-specific adaptations − such as dynamic $\lambda$ scheduling based on value convergence rates or trajectory-level uncertainty − represent promising directions for future work.

## Acknowledgements

The work is supported by the National Natural Science Foundation of China (No. U21A20473, No. 62276160), the Fundamental Research Program of Shanxi Province (No. 202503021212091, No. 202403021222153), the Shanxi Provincial Natural Science Foundation General Project (No. 202203021211294), the Shanxi Provincial Overseas Study Fund Project (No. 20240002), the Scientific and Technological Innovation Programs of Higher Education Institutions in Shanxi (No. 2025L001).

## Impact Statement

This paper presents work whose goal is to advance the field of Machine Learning. There are many potential societal consequences of our work, none which we feel must be specifically highlighted here.

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

# A. Algorithms and Details

In this section, we provide detailed algorithms of SAM, SAM with Weighted Sharpness, SAM with Weighted Samples, and the proposed Q-SAM.

---

**Algorithm 2** Sharpness-Aware Minimization (SAM)

---

**Input:** Training set $\mathcal{S} \triangleq \cup_{i=1}^{n}\{(x_i, y_i)\}$, loss function $\mathcal{L}(\phi)$, batch size $b$, step size $\alpha$, perturbation bound $\rho > 0$.

**Output:** Model trained with SAM

Initialize weights $\phi_0$, $k = 0$;

**while** not converged **do**

    Sample batch $\mathcal{B} = \{(x_1, y_1), \ldots, (x_b, y_b)\}$;

    Compute gradient $g_k^{Adam} = \nabla_\phi \mathcal{L}_{\mathcal{B}}(\phi)$ of the batch's training loss;

    Compute $\epsilon^*$ per Equation (2);

    Compute gradient approximation for the SAM objective (Equation (4)): $g_k^{SAM} = \nabla_\phi \mathcal{L}_{\mathcal{B}}(\phi)|_{\phi+\epsilon^*}$;

    Update weights: $\phi_{k+1} = \phi_k - \alpha g_k^{SAM}$;

**end while**

Return $\phi_k$

---

**Algorithm 3** SAM with Weighted Sharpness

---

**Input:** Training set $\mathcal{S} \triangleq \cup_{i=1}^{n}\{(x_i, y_i)\}$, loss function $\mathcal{L}(\phi)$, batch size $b$, step size $\alpha$, perturbation bound $\rho > 0$, parameter $\lambda \in [0, 1)$.

**Output:** Model trained with SAM

Initialize weights $\phi_0$, $k = 0$;

**while** not converged **do**

    Sample batch $\mathcal{B} = \{(x_1, y_1), \ldots, (x_b, y_b)\}$;

    Compute gradient $g_t^{Adam} = \nabla_\phi \mathcal{L}_{\mathcal{B}}(\phi)$ of the batch's training loss;

    Compute $\epsilon^*$ per Equation (2);

    Compute gradient approximation for the SAM objective (Equation (4)): $g_k^{SAM} = \nabla_\phi \mathcal{L}_{\mathcal{B}}(\phi)|_{\phi+\epsilon^*}$;

    Update weights: $\phi_{k+1} = \phi_k - \alpha(\frac{\lambda}{1-\lambda} g_k^{SAM} + \frac{1-2\lambda}{1-\lambda} g_k^{Adam})$;

**end while**

Return $\phi_k$

---

---

**Algorithm 4** SAM with Weighted Samples: Does not include the SAM optimization process

---

**Input:** Initialize Q-network $Q_\theta(s, a)$ and an error model $\Delta_\varphi(s, a)$, batch size $b$.

**for** iteration $k$ in $\{1, \cdots, N\}$ **do**

    Sample batch $\mathcal{B} = \{(s_i, a_i)\}_{i=1}^b$;

    Evaluate $Q_\theta(s, a)$ and $\Delta_\varphi(s, a)$ on samples $(s_i, a_i)$;

    Compute target values for $Q$ and $\Delta$ on samples:

    $y_i = r_i + \gamma \max_{a'} Q_{k-1}(s_i', a')$,

    $\hat{a}_i = \arg\max_a Q_{k-1}(s_i', a)$,

    $\hat{\Delta}_i = \big|Q_\theta(s, a) - y_i\big| + \gamma \Delta_{k-1}(s_i', \hat{a}_i)$;

    Compute $w_k$ using Equation (7);

    Minimize Bellman error for $Q_\theta$ weighted by $w_k$:

    $\theta_{k+1} \leftarrow \underset{\theta}{\arg\min} \frac{1}{N} \sum_i^N w_k(s_i, a_i)(Q_\theta(s_i, a_i) - y_i)^2$;

    Minimize model error for training $\varphi$:

    $\varphi_{k+1} \leftarrow \underset{\varphi}{\arg\min} \frac{1}{N} \sum_{i=1}^N (\Delta_\varphi(s_i, a_i) - \hat{\Delta}_i)^2$

**end for**

---

**Algorithm 5** Q bound weighted Sharpness-Aware Minimization (Q-SAM): Actor-Critic Version

---

**Input:** Offline training dataset $\mathcal{D}$, Initialize Q-network $Q_\theta(s, a)$ and an error model $\Delta_\varphi(s, a)$, batch size $b$, step size $\alpha$, perturbation bound $\rho > 0$, parameter $\lambda \in [0, 1)$.

**while** not converged **do**

    Sample batch $\mathcal{B} = \{(s_i, a_i)\}_{i=1}^b$;

    # SAM with Weighted Samples

    Compute sample weights and update the Critic using Algorithm 4;

    # SAM with Weighted Sharpness

    Update the Actor using Algorithm 3;

**end while**

# B. Experimental Details and Additional Results

## B.1. Experimental Details

Datasets in Gym MuJoCo:

- medium: Train to halfway using online Soft Actor-Critic and then collect 1M samples using this policy.

- medium-replay: The data collected by the entire replay buffer when trained to a medium quality level.

- medium-expert: Equivalent mixed expert datasets and suboptimal datasets, with the suboptimal data obtained by either a suboptimal policy or a randomized policy.

Datasets in AntMaze:

The AntMaze domain utilizes the same umaze, medium, and large maze layouts as the Maze2D domain but replaces the agent with the "Ant" robot from the OpenAI Gym MuJoCo benchmark. The dataset for 'antmaze-umaze' is collected by directing the agent to a fixed goal location from a fixed starting position, which are situated on opposite sides of the wall in the umaze.

For more challenging tasks, the "diverse" dataset is created by randomly selecting goal locations within the maze and guiding the ant toward them. In contrast, the "play" dataset is curated by selecting specific goal locations paired with carefully chosen initial positions.

**Implementation details.** All of the baseline algorithms come from the code library $\mathcal{CORL}$: https://github.com/tinkoff-ai/CORL. We are preparing to release a new open-source codebase, which will enable users to implement the SAM optimizer as required for their specific use cases. All experiments run on a server equipped with an Intel® Xeon® Gold 6254 CPU @ 3.10GHz and NVIDIA GeForce RTX 3090 GPU.

## B.2. Results Reported with Full Standard Deviations

*Table 6.* Average normalized scores over five seeds. We **bold** the highest mean.

| Dataset | TD3BC | TD3BC + SAM | TD3BC + Q-SAM | CQL | CQL + SAM | CQL + Q-SAM | IQL | IQL + SAM | IQL + Q-SAM |
|---|---|---|---|---|---|---|---|---|---|
| ha-m | 48.1±0.2 | **48.5**±0.3 | 48.4±0.4 | 47.0±0.2 | 47.1±0.3 | **47.3**±0.2 | 48.3±0.2 | 48.4±0.2 | **49.2**±0.1 |
| ha-mr | 44.8±0.6 | 44.6±0.4 | **45.3**±0.4 | 45.0±0.3 | 45.5±0.3 | **45.6**±0.2 | 44.5±0.2 | 44.6±0.4 | **45.1**±0.2 |
| ha-me | 90.8±6.0 | 90.3±5.2 | **93.9**±4.6 | 95.6±0.4 | 94.1±1.2 | 95.6±0.5 | 94.7±0.5 | 95.0±0.4 | **95.9**±0.4 |
| ho-m | 60.4±3.5 | 60.9±3.2 | **65.7**±2.6 | 59.1±3.8 | 59.7±2.2 | 70.3±2.5 | 67.5±3.8 | 68.5±2.8 | **75.3**±2.2 |
| ho-mr | 64.4±21.5 | 69.4±12.5 | **87.3**±5.0 | 95.1±5.3 | 94.9±4.7 | 95.5±5.6 | 97.4±6.4 | 100.5±5.8 | **103.0**±5.5 |
| ho-me | 101.2±9.1 | 101.5±9.0 | **105.1**±8.4 | 99.3±10.9 | 108.7±4.4 | 109.8±1.7 | 107.4±7.8 | 109.4±5.7 | **113.4**±5.1 |
| wa-m | 82.7±4.8 | 82.9±4.4 | **85.4**±1.0 | 80.8±3.3 | 81.7±1.2 | 84.2±1.2 | 80.9±3.2 | 86.9±3.5 | **87.6**±3.0 |
| wa-mr | 85.6±4.0 | 86.7±3.5 | **90.1**±1.0 | 73.1±13.2 | 74.9±9.0 | 84.4±4.2 | 82.2±3.0 | 87.2±2.0 | **90.9**±4.7 |
| wa-me | 110.0±0.4 | 109.0±0.2 | **111.5**±0.2 | 109.5±0.4 | 109.6±0.4 | 112.3±0.3 | 111.7±0.9 | 111.3±0.6 | **111.7**±1.0 |
| Total | 688.0 | 693.8 | **732.7** | 704.5 | 716.2 | **745.0** | 734.6 | 751.8 | **772.1** |
| ant-u | 70.8±39.2 | 80.3±12.8 | **92.8**±2.5 | 92.8±1.9 | 92.3±2.3 | 93.3±3.3 | 77.0±5.5 | 77.5±4.7 | **81.3**±6.8 |
| ant-ud | 44.8±11.6 | 52.8±13.5 | **66.3**±15.5 | 37.3±3.7 | 36.3±3.2 | 38.3±4.1 | 54.3±5.5 | 56.7±5.3 | **65.9**±6.3 |
| ant-mp | 0.3±0.4 | 0.3±0.5 | **3.8**±0.9 | 65.8±11.6 | 68.5±10.1 | 71.8±7.6 | 65.8±11.8 | 68.5±9.2 | **80.8**±8.5 |
| ant-md | 0.3±0.4 | 0.3±0.3 | **0.5**±0.3 | 67.3±3.6 | 68.1±2.6 | 68.5±2.5 | 73.8±5.5 | 75.9±5.1 | **81.2**±5.5 |
| ant-lp | 0.0±0.0 | 0.0±0.0 | 0.0±0.0 | 20.8±7.3 | 28.0±5.5 | 33.8±6.2 | 42.0±4.5 | 48.5±4.2 | **63.3**±7.6 |
| ant-ld | 0.0±0.0 | 0.0±0.0 | 0.0±0.0 | 20.5±13.2 | 23.3±6.7 | 29.3±6.4 | 30.3±3.6 | 30.5±3.2 | **30.8**±3.9 |
| Total | 116.2 | 133.7 | **163.4** | 304.5 | 316.5 | **335.0** | 343.2 | 357.6 | **403.3** |

## B.3. Complete Small-sample Experimental Results

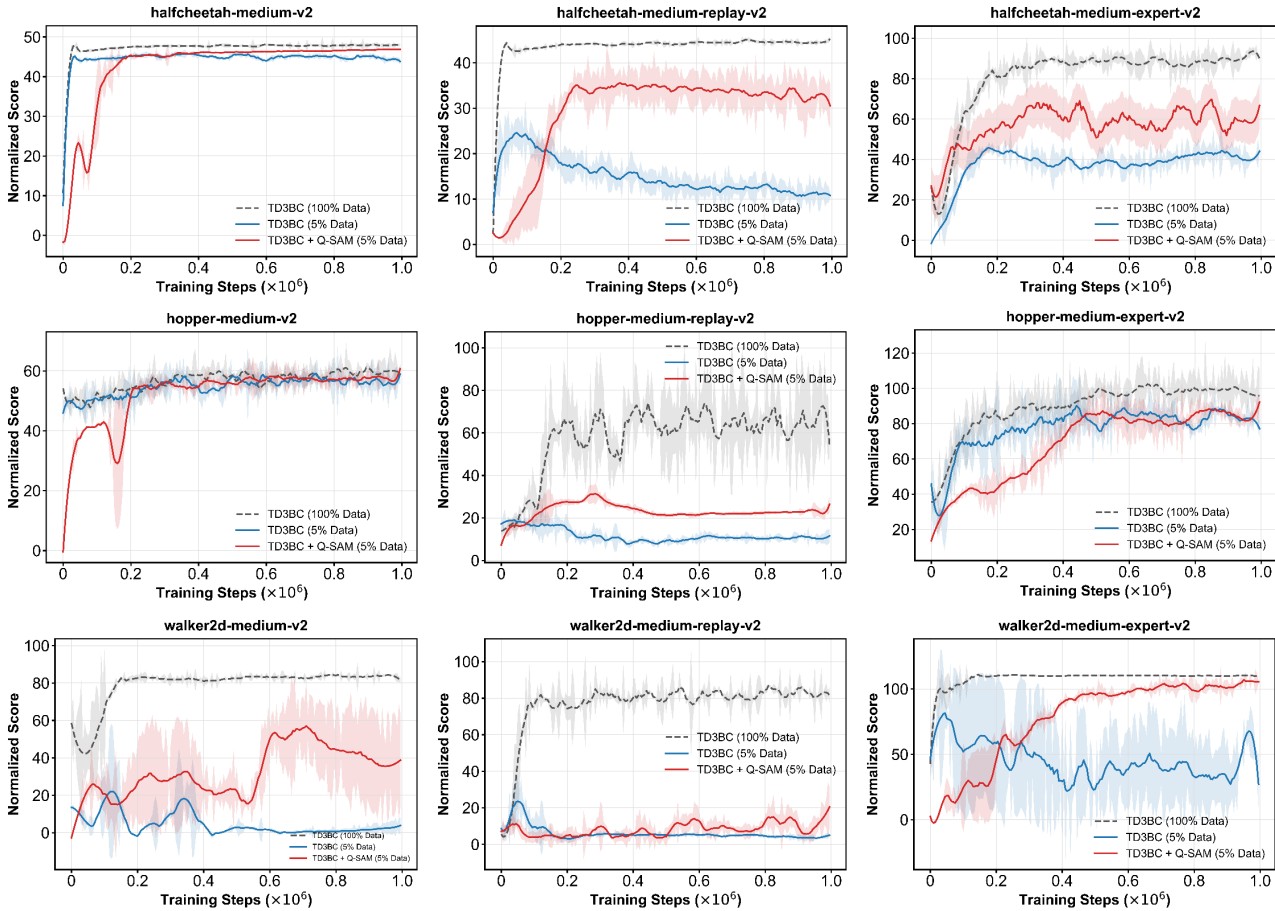

*Figure 6.* The small-sample performance of different algorithms.

## B.4. Complete Ablation Results

*Table 7.* Average normalized scores over five seeds. "Drop" means the decrease in mean values.

| Dataset | TD3BC + Q-SAM | TD3BC + Q-SAM w/o Weighted Sharpness | | TD3BC + Q-SAM w/o Weighted Samples | |
|---|---|---|---|---|---|
| | | Score | Drop | Score | Drop |
| halfcheetah-m | 48.4±0.4 | 48.3±0.2 | 0.1 | 48.6±0.4 | -0.2 |
| halfcheetah-mr | 45.3±0.4 | 45.0±0.4 | 0.3 | 45.2±0.4 | 0.1 |
| halfcheetah-me | 93.9±4.6 | 91.6±5.3 | 2.3 | 93.3±4.0 | 0.6 |
| hopper-m | 65.7±2.6 | 62.7±3.2 | 3.0 | 63.7±2.0 | 2.0 |
| hopper-mr | 87.3±5.0 | 74.5±8.9 | 12.8 | 87.4±3.9 | -0.1 |
| hopper-me | 105.1±8.4 | 101.9±8.7 | 3.2 | 104.9±8.2 | 0.2 |
| walker2d-m | 85.4±1.0 | 83.4±3.8 | 2.0 | 85.5±0.8 | -0.1 |
| walker2d-mr | 90.1±1.0 | 88.9±2.8 | 1.2 | 89.8±1.1 | 0.3 |
| walker2d-me | 111.5±0.2 | 109.2±0.2 | 2.3 | 110.5±0.3 | 1.0 |
| Total | 732.7±23.6 | 705.5±33.5 | *27.2* | 728.9±13.5 | *3.8* |
| antmaze-u | 92.8±2.5 | 90.5±2.6 | 2.3 | 92.3±2.3 | 0.5 |
| antmaze-ud | 66.3±15.5 | 55.2±14.5 | 11.1 | 65.0±15.7 | 1.3 |
| antmaze-mp | 3.8±0.9 | 2.9±1.9 | 0.9 | 2.8±2.3 | 1.0 |
| antmaze-md | 0.5±0.3 | 0.4±0.3 | 0.1 | 0.5±0.5 | 0.0 |
| antmaze-lp | 0.0±0.0 | 0.0±0.0 | 0.0 | 0.0±0.0 | 0.0 |
| antmaze-ld | 0.0±0.0 | 0.0±0.0 | 0.0 | 0.0±0.0 | 0.0 |
| Total | 163.4±19.2 | 149.0±19.3 | *14.4* | 160.6±20.8 | *2.8* |

*Table 8.* Average normalized scores over five seeds. "Drop" means the decrease in mean values.

| Dataset | IQL + Q-SAM | IQL + Q-SAM w/o Weighted Sharpness | | IQL + Q-SAM w/o Weighted Samples | |
|---|---|---|---|---|---|
| | | Score | Drop | Score | Drop |
| halfcheetah-m | 49.2±0.1 | 48.6±0.2 | 0.6 | 48.9±0.5 | 0.3 |
| halfcheetah-mr | 45.1±0.2 | 44.7±0.4 | 0.4 | 44.9±0.7 | 0.2 |
| halfcheetah-me | 95.9±0.4 | 94.9±0.4 | 1.0 | 95.5±0.5 | 0.4 |
| hopper-m | 75.3±2.2 | 70.3±2.8 | 5.0 | 73.1±2.4 | 2.2 |
| hopper-mr | 103.0±5.5 | 100.2±6.1 | 2.8 | 101.7±6.3 | 1.3 |
| hopper-me | 113.4±5.1 | 111.3±5.5 | 2.1 | 113.2±5.2 | 0.2 |
| walker2d-m | 87.6±3.0 | 82.4±3.2 | 5.2 | 87.1±3.2 | 0.5 |
| walker2d-mr | 90.9±4.7 | 84.5±4.3 | 6.4 | 89.1±6.3 | 1.8 |
| walker2d-me | 111.7±1.0 | 111.6±0.8 | 0.1 | 112.5±0.8 | -0.8 |
| Total | 772.1±22.2 | 748.5±23.7 | *23.6* | 766.0±27.9 | *6.1* |
| antmaze-u | 81.3±6.8 | 79.8±6.3 | 1.5 | 80.0±6.1 | 1.3 |
| antmaze-ud | 65.9±6.3 | 55.8±6.4 | 10.1 | 64.3±5.6 | 1.6 |
| antmaze-mp | 80.8±8.5 | 66.8±10.9 | 14.0 | 80.4±8.2 | 0.4 |
| antmaze-md | 81.2±5.5 | 74.2±5.8 | 7.0 | 80.1±5.8 | 1.1 |
| antmaze-lp | 63.3±7.6 | 46.6±8.3 | 16.7 | 60.1±8.2 | 3.2 |
| antmaze-ld | 30.8±3.9 | 30.3±3.8 | 0.5 | 30.6±4.2 | 0.2 |
| Total | 403.3±38.6 | 353.5±41.5 | *49.8* | 395.5±38.1 | *7.8* |

*Table 9.* Average normalized scores over five seeds. "Drop" means the decrease in mean values.

| Dataset | CQL + Q-SAM | CQL + Q-SAM w/o Weighted Sharpness | | CQL + Q-SAM w/o Weighted Samples | |
|---|---|---|---|---|---|
| | | Score | Drop | Score | Drop |
| antmaze-u | 93.3±3.3 | 92.8±2.8 | 0.5 | 93.0±3.0 | 0.3 |
| antmaze-ud | 38.3±4.1 | 37.3±3.7 | 1.0 | 37.3±4.5 | 1.0 |
| antmaze-mp | 71.8±7.6 | 69.0±9.4 | 2.8 | 70.5±7.0 | 1.3 |
| antmaze-md | 68.5±2.5 | 67.8±2.3 | 0.7 | 68.5±2.3 | 0.0 |
| antmaze-lp | 33.8±6.2 | 25.8±6.3 | 8.0 | 32.8±6.5 | 1.0 |
| antmaze-ld | 29.3±6.4 | 22.2±8.6 | 7.1 | 28.5±8.3 | 0.8 |
| Total | 335.0±38.6 | 314.9±33.1 | *20.1* | 330.6±31.6 | *4.4* |

*Table 10.* Performance under different parameters $\lambda$. The $\lambda$ comes from WSAM in Q-SAM, and its use generally leads to performance improvements compared to using SAM alone. "Average" indicates that using CQL and IQL as a baseline can share parameters, while TD3BC uses a different set.

| Algorithm | $\lambda$ | halfcheetah | | | hopper | | | walker2d | | | Average | Proportion |
|---|---|---|---|---|---|---|---|---|---|---|---|---|
| | | -m | -mr | -me | -m | -mr | -me | -m | -mr | -me | | |
| TD3BC +Q-SAM | $\lambda = 0.1$ | 48.4 | **45.3** | **93.9** | 63.7 | 76.8 | **105.1** | 84.2 | 87.5 | **111.5** | 79.6 | 5/9 |
| | $\lambda = 0.9$ | 48.2 | 45.0 | 91.7 | **65.7** | **87.3** | 104.6 | **85.4** | **90.1** | 111.2 | **81.0** | 4/9 |
| TD3BC +SAM | – | **48.5** | 44.6 | 90.3 | 60.9 | 69.4 | 101.5 | 82.9 | 86.7 | 109.0 | 77.1 | – |
| CQL +Q-SAM | $\lambda = 0.1$ | **47.3** | **45.6** | **95.6** | **70.3** | **95.5** | 109.5 | **84.2** | **84.4** | 111.8 | **82.7** | 7/9 |
| | $\lambda = 0.9$ | 47.1 | 45.5 | 94.9 | 65.5 | 95.3 | **109.8** | 82.8 | 80.7 | **112.3** | 81.5 | 2/9 |
| CQL +SAM | – | 47.1 | 45.5 | 94.1 | 59.7 | 94.9 | 108.7 | 81.7 | 74.9 | 109.6 | 79.6 | – |
| IQL +Q-SAM | $\lambda = 0.1$ | **49.2** | **45.1** | **95.9** | **75.3** | **103.0** | 111.9 | **87.6** | 88.3 | **111.7** | **85.3** | 7/9 |
| | $\lambda = 0.9$ | 48.8 | 44.6 | 95.5 | 70.9 | 102.1 | **113.4** | 87.4 | **90.9** | 111.6 | 85.0 | 2/9 |
| IQL +SAM | – | 48.4 | 44.6 | 95.0 | 68.5 | 100.5 | 109.4 | 86.9 | 87.2 | 111.3 | 83.5 | – |

## B.5. Computation of $w_k$ and Selection of $\lambda$

In practice, the computation of $w_k$ follows a **five-step process**:

1. **Estimate the next-step cumulative error $\Delta_{k-1}$.** Query the target error network to obtain the cumulative Bellman error $\Delta_{k-1}(s', a')$ for the next state-action pair. In implementations with twin critics, this query is performed for both Q-networks.

2. **Compute exponential weights (accounting for terminal states and temperature scaling).** Calculate the scaled term $-\gamma \cdot \Delta_{k-1}(s', a')/\tau$, where a terminal-state mask $(1 - d)$ zeros out the contribution of terminal transitions to prevent error propagation beyond episode boundaries, and $\tau$ denotes the temperature hyperparameter.

3. **Self-normalization (Softmax) to obtain $w_k$.** Apply batch-wise Softmax normalization to ensure the weights sum to unity:

$$w_k = \frac{\exp(-\gamma\Delta_{k-1}(s', a')/\tau)}{\sum_j \exp(-\gamma\Delta_{k-1}(s'_j, a'_j)/\tau)}. \tag{9}$$

4. **Compute cumulative error targets (for training the error network).** Combine the current Bellman residual with the discounted next-step error to form the supervised target for updating the error network:

$$\Delta_k^{\text{target}} = \left| Q_k(s, a) - \mathcal{T}Q_{k-1}(s, a) \right| + \gamma(1 - d)\Delta_{k-1}(s', a'). \tag{10}$$

5. **Apply $w_k$ to the critic loss.** Reweight the squared temporal-difference loss using the derived importance weights:

$$\mathcal{L}_{\text{critic}} = \mathbb{E}_{(s,a)\sim\mathcal{D}}\left[w_k \cdot \left(Q(s,a) - Q^{\text{target}}(s,a)\right)^2\right]. \tag{11}$$

The only parameter in our proposed method, Q-SAM, is the $\lambda$ in the *Weighted Sharpness* component, which scales the sharpness term and influences the gradient direction computed under perturbations. Due to the varying nature of tasks and data, finding the optimal $\lambda$ for each task to achieve the best performance is time-consuming. However, our goal is not solely to maximize performance, but to demonstrate that *Weighted Sharpness* itself can improve performance. More importantly, we aim to explore how the two distinct directions guided by $\lambda$ affect the algorithm's behavior. In Table 10, we show the performance preferences of all related methods for two extreme values of $\lambda$, 0.1 and 0.9, where a check mark indicates better performance under a given parameter. It can be observed that $\lambda = 0.1$ generally leads to more performance benefits, suggesting that in offline RL, SAM should be more conservative to reduce the impact on minimizing the minima, thus avoiding issues like extrapolation and Bellman errors. However, some tasks benefit from $\lambda = 0.9$, which may help escape local optima in certain cases. Therefore, the role of $\lambda$ is to strike a balance in offline RL between minimizing extrapolation errors and avoiding local optima. This highlights the potential for future research on adaptive $\lambda$, which is an intriguing direction to explore.

### B.6. Related Work

**Offline RL and Generalization Challenges.** Offline RL has increasingly become a prominent subfield within reinforcement learning, with its core challenge lying in achieving robust generalization to unseen distributions despite the inherent limitations of static datasets. Existing approaches often address OOD data by employing techniques such as behavior policy constraints (Fujimoto & Gu, 2021; Fakoor et al., 2021; Wu et al., 2022; Li et al., 2023b; Ran et al., 2023), value function regularization (Kumar et al., 2020b; Kostrikov et al., 2021; Ma et al., 2021), uncertainty estimation (An et al., 2021; Bai et al., 2022; Wu et al., 2021; Wang et al., 2025b; Qiao et al., 2025), and density ratio estimation (Lee et al., 2022; Sikchi et al., 2024; Mao et al., 2024b;a). However, these methods primarily focus on data utilization and bias correction, while paying limited attention to structurally enhancing generalization capabilities.

In response, recent efforts have explored novel strategies to improve generalization to unknown data, such as relaxing constraints (Lyu et al., 2022; Ran et al., 2023; Wu et al., 2022; Mao et al., 2023) and regarding the generalization challenge as a domain adaptation problem (Qi et al., 2022; Wang et al., 2024; 2025a). Breaking through the inherent limitations of existing algorithms on static datasets has thus emerged as a frontier research question in this field. In contrast, our approach takes a unique perspective by focusing on the choice of optimizer, a previously underexplored aspect of offline RL. By leveraging this new theoretical tool to reshape generalization performance, our method has the potential to open a novel addition for offline RL research.

**SAM in Deep Learning.** SAM is a widely adopted optimization method in deep learning fields such as CV and NLP, achieving remarkable results across various tasks in recent years. Numerous SAM variants have been developed to enhance effectiveness (Zhuang et al., 2022; Kim et al., 2022; Yue et al., 2023), efficiency (Du et al., 2022), and generalization performance (Kwon et al., 2021; Wang et al., 2023). Rather than focusing on which variant delivers state-of-the-art results, we prioritize selecting the most suitable method based on the specific challenges SAM faces in offline RL.

