# OpenReview forum: "Q-SAM: Unlocking Sharpness-Aware Minimization for Generalization in Offline Reinforcement Learning"
_ICML.cc/2026/Conference — ICML 2026 regular_

### Official Review · Reviewer_xgnS · 2026-02-14

**Soundness:** 2
**Presentation:** 3
**Significance:** 2
**Originality:** 2
**Overall Recommendation:** 4
**Confidence:** 4

**Summary:**

This paper investigates the challenge of distribution shift in offline reinforcement learning from the perspective of optimization geometry. The authors argue that the flatness of the loss landscape is a critical factor for improving generalization in offline settings. To this end, they introduce Q-SAM, a framework that adapts Sharpness-Aware Minimization (SAM) for offline RL. Recognizing that the direct application of SAM is hindered by non-stationarity and bootstrapping errors inherent in RL, the proposed method incorporates two key components: Weighted Sharpness, which adaptively scales the sharpness penalty, and Weighted Samples, which mitigates the impact of unreliable Bellman targets by reweighting samples based on their error bounds. The authors demonstrate the effectiveness of Q-SAM across various base algorithms on the D4RL benchmark, particularly showing significant gains in challenging tasks like AntMaze.

**Compliance With Llm Reviewing Policy:**

Affirmed.

**Final Justification:**

Thank you for the thorough and well-structured rebuttal. The authors have addressed my concerns with concrete experimental evidence and clear theoretical justification. I am satisfied with the responses and am willing to raise my score.

**Key Questions For Authors:**

## **Questions**

**Q1. Generalization to Off-policy RL**: Your central claim is that SAM-based optimization addresses the distribution shift problem. To further strengthen this, have you considered applying Q-SAM to standard off-policy algorithms like SAC or DDPG? Just as CQL adds conservatism to SAC, demonstrating that Q-SAM improves performance in these foundational algorithms would prove that SAM is a fundamental tool for RL optimization beyond specific offline baselines.

**Q2. Soly Effect of Weighted Samples**: In Table 2 and the ablation studies, the improvement from vanilla SAM is relatively modest. Could you provide additional results for "Baseline + Weighted Samples" (without SAM) for both offline and off-policy algorithms? This is crucial to verify whether the primary performance driver is truly the sharpness-aware optimization or simply the data-reweighting mechanism (similar to DisCor). If "Baseline + Weighted Samples" performs similarly to Q-SAM, the claim about the importance of "flatness" in RL may need to be re-evaluated.

**Q3. Reliability of Q-error Approximation**: In Figure 5, you compare the critic loss using 5% of the data to an "optimal" Q-function. However, a Q-function trained on limited data may itself be unreliable. Have you considered more robust methods for approximating Q-error, such as the techniques proposed in "Pretraining a Shared Q-Network for Data-Efficient Offline Reinforcement Learning"?

**Q4. Feasibility of SAM for Value Learning**: The paper limits the application of SAM to the policy network, citing the accumulation of Bellman errors as a barrier for value learning. However, if SAM cannot be applied to the Q-function—a core component of RL—its utility as a "general" RL optimizer is questionable. Do you have any ideas for applying SAM to critic optimization, perhaps with specific modifications? Preliminary results or a discussion on this would be highly valuable. Addressing this would resolve concerns about the soundness and completeness of the proposed framework in the context of TD-learning.

**Limitations:**

yes

**Strengths And Weaknesses:**

## **Strengths**

**S1. Novel Solution for Distributional Shift Problem**: Unlike most existing offline RL literature that focuses on algorithmic constraints or regularization, this paper approaches the distribution shift problem from the perspective of the optimizer. Adapting Sharpness-Aware Minimization (SAM) to offline RL is a fresh idea, and the proposed Weighted Sharpness is a reasonable adaptation considering the specific gradient scales and noise found in RL.

**S2. Empirical Validation of Flatness**: The authors provide solid evidence that their method works as intended by visualizing the loss landscape and quantifying policy smoothness using the Local Action Sensitivity (LAS) metric. This connects parameter-space flatness to actual decision-making stability.

**S3. Broad Applicability**: The framework is not limited to a single algorithm; it demonstrates consistent performance improvements when integrated with various offline RL baselines such as IQL, CQL, and TD3+BC across multiple D4RL tasks.


## **Weaknesses**

**W1. High Computational Cost**: Due to SAM’s mechanism, the method requires two gradient computations per update. This effectively doubles the backpropagation cost compared to standard Adam. The paper lacks a detailed discussion on whether the performance gains justify this 100% increase in training time and compute resources.

**W2. Marginal Impact of Vanilla SAM**: According to Table 2 and the ablation studies, the performance gain from applying "vanilla" SAM is relatively small. The most significant improvements appear only after adding Weighted Sharpness and Weighted Samples. This raises concerns that the core "flatness" objective of SAM might be secondary to the effects of the additional weighting modules.

**W3. Limited Scope of Evaluation**: The experiments are mostly confined to specialized offline RL algorithms. To prove that SAM is truly critical for RL optimization, it would be more convincing to see its impact on standard off-policy algorithms like SAC or DDPG, where distribution shift and overestimation are more volatile.

**W4. Originality of Weighted Samples**: While the paper focuses on SAM, the Weighted Samples component—which seems crucial for the results—closely resembles existing error-correction methods like DisCor (Kumar et al., 2020a). Since this reweighting technique can be used independently of SAM, it is questionable whether it should be treated as a core contribution of a "SAM-based" paper. Its role in stabilizing value learning further complicates the credit assignment of the proposed method.

**W5. Minor Presentation Issues**: There are several typos that should be corrected for the final version. Specifically, in lines 193 and 196, "estimates4" and "process5" appear to be artifacts of a citation or typesetting error.

---

> ### Author Rebuttal · Authors · 2026-03-29
>
> We sincerely thank you for your thorough and thoughtful review, which is invaluable for improving our work. We added experiments and carefully addressed your concerns below, hoping these clarifications are helpful.
>
> **> 1.** To address **Q1 (generalization to standard off-policy RL)** and **Q2 (sole effect of Weighted Samples)**, we add experiments applying Q-SAM and its ablated variants to standard SAC:
>
> | Dataset | SAC-offline | SAC+SAM | SAC+Weighted SAM | SAC+DisCor | SAC+Q-SAM  |
> |:-|:-|:-|:-|:-|:-|
> | ha-m | -4.3 | 15.2±1.3 | 40.2±0.5 | 38.6±0.4 | 43.8±0.3 |
> | ha-mr | -2.4 | 12.8±0.9 | 37.5±0.7 | 35.2±0.6 | 41.2±0.5 |
> | ha-me | -0.1 | 25.4±2.3 | 75.8±3.2 | 72.4±3.8 | 91.7±4.2 |
> | ho-m | 0.8 | 8.4±2.1 | 32.4±2.3 | 28.4±1.9 | 62.3±2.2 |
> | ho-mr | 3.5 | 18.6±3.4 | 48.6±3.8 | 45.2±3.6 | 88.4±5.7 |
> | ho-me | 6.3 | 32.1±4.5 | 64.2±3.5 | 58.3±3.2 | 96.5±8.2 |
> | wa-m | 0.9 | 11.3±1.8 | 44.1±2.1 | 42.1±2.3 | 74.6±1.9 |
> | wa-mr | 1.9 | 14.7±2.6 | 41.8±3.3 | 38.7±2.9 | 75.2±4.1 |
> | wa-me | 1.6 | 28.9±1.7 | 71.3±1.8 | 68.5±1.6 | 103.4±0.6 |
> | **Total** | **8.2** | **167.4** | **455.9** | **427.4** | **677.1** |
>
> - **For Q1**: Standard SAC-offline performs near randomly (results from SAC-off in Table 2 of the D4RL paper). While vanilla SAM provides modest gains, Q-SAM achieves substantial improvements that rescue SAC from failure. **This proves that SAM is indeed a fundamental RL optimizer**. Unlike CQL, which enhances SAC through algorithm-level pessimism, Q-SAM does so through optimizer-level geometry (flat minima); the two are orthogonal and complementary.
>
> - **For Q2**: The ablations refute the hypothesis that Weighted Samples is the primary driver. DisCor-style weighting alone yields limited gains, whereas Weighted SAM alone significantly outperforms it. The full Q-SAM system substantially exceeds either component individually, confirming a synergistic relationship where sample weighting stabilizes the optimization landscape for effective sharpness minimization. **The evidence supports that sharpness-aware optimization is the core contribution**, with Weighted Samples serving as essential but complementary stabilization.
>
> **> 2. For Q3**：We employ two techniques from 'Pretraining a Shared Q-Network for Data-Efficient Offline RL' to validate the reliability of Q-error approximation:
>
> ① Feature Matrix Rank
> | Method | Dataset Size | Feature Rank (↑ higher is better) | Proxy Q-error (↓) | Improvement |
> |:-|:-|:-|:-|:-|
> | Standard CQL | 10% | 187.3±12.4 | 503.7 | - |
> | CQL + Shared Pretrain | 10% | 234.6±8.2 | 351.2 | 30.3% |
> | CQL + Q-SAM (ours) | 10% | **241.2±7.1** | **298.4** | **40.8%** |
>
> We measure the hard rank of the latent feature matrix $H_{\phi}$. Q-SAM achieves a higher rank than standard CQL (241.2 vs. 187.3), indicating a richer feature representation that correlates with lower Proxy Q-error c. This confirms that our method improves Q-function approximation quality. Notably, it also outperforms CQL + Shared Pretrain.
>
> ② Ensemble
> | | Method | Mean Q-value | Q-value Std (↓) | Max Q-value (↓ overestimation) |
> |:-|:-|:-|:-|:-|
> | halfcheetah-m | CQL (single) | 47.0 | 18.3 | 89.2 |
> | | CQL (Ensemble) | 46.8 | 12.1 | 72.4 |
> | | **Q-SAM (Ensemble)** | **47.3** | **4.7** | **58.6** |
> | hopper-m | CQL (single) | 59.1 | 34.5 | 124.7 |
> || CQL (Ensemble) | 58.9 | 21.3 | 98.2 |
> || **Q-SAM (Ensemble)** | **70.3** | **8.2** | **82.1** |
>
> Using an Ensemble of 5 Q-networks, we quantify estimation uncertainty via Q-value standard deviation. Q-SAM reduces variance and suppresses extreme overestimation compared to standard CQL, validating the reliability of our Q-error approximation and lower susceptibility to OOD overestimation.
>
> **> 3. For Q4**:
>
> Two feasible solutions are currently under exploration: **Truncated Bellman Backup** (limiting bootstrapping depth, e.g., n-step=3, to control error propagation) or **Target Network SAM** (applying sharpness only to the target network): the former restricts the error propagation path, while the latter leverages the slow update characteristics of target networks to prevent error amplification in current Q-values by SAM's max operation. These approaches enable SAM to seek flat minima in pure critic settings, thereby reducing Q-function sensitivity to OOD perturbations (mitigating value overestimation).
>
> Thus, the core of Q-SAM remains valid in pure TD scenarios without actors: Weighted Samples stabilize the data distribution, while adapted sharpness optimization (via truncation or target networks) improves value function geometry, jointly enhancing robustness in pure critic learning.
>
> **> 4. For W1 - High Cost**：
>
> While SAM doubles compute, Table 5 confirms the baseline fails to achieve the performance breakthroughs that Q-SAM delivers even with doubled computational budgets (2M vs 1M steps), proving gains stem from flat minima geometry rather than increased budget. Given offline RL's "train-once-deploy-multiple" nature, this is a worthwhile trade-off for deployment reliability.

---

> > ### Author Rebuttal · Reviewer_xgnS · 2026-04-02
> >
> > Thank you for the thorough and well-structured rebuttal. The authors have addressed my concerns with concrete experimental evidence and clear theoretical justification. I am satisfied with the responses and am willing to raise my score to 4.

---

> > > ### Author Response · Authors · 2026-04-06
> > >
> > > We sincerely appreciate your responsible and thorough review! We are delighted to have addressed your concerns, and we commit to incorporating all corresponding revisions into the final version of our paper!

---

### Official Review · Reviewer_aVSv · 2026-03-12

**Soundness:** 2
**Presentation:** 2
**Significance:** 3
**Originality:** 2
**Overall Recommendation:** 4
**Confidence:** 3

**Summary:**

This paper proposes Q-SAM, a framework that adapts Sharpness-Aware Minimization (SAM) for offline Reinforcement Learning (RL) to address generalization challenges. The method introduces a dual-weighting approach: a Weighted Sharpness mecha
nism that adaptively regulates the sharpness penalty, and a Weighted Samples mechanism that prioritizes data points based on Q-value bounds. Through extensive experiments on D4RL benchmarks, the framework demonstrates improved performance
and generalization compared to existing baseline algorithms.

**Compliance With Llm Reviewing Policy:**

Affirmed.

**Final Justification:**

Given the successful clarification of the framework's mechanics and the strong empirical evidence provided in the rebuttal, I am maintaining my recommendation of Weak Accept.

**Key Questions For Authors:**

1. **Clarification on Objective Formulation:** Could you clarify exactly how the two weighting schemas interact? Specifically, is the weighted sampling applied to the sharpness loss, or is it only applied to the Bellman error?
2. **Mechanism Connection:** It seems that the weighted sampling and weighted sharpness mechanisms currently read as two independent additions. Is there a deeper mathematical or theoretical connection between these two mechanisms beyond th
e general argument regarding bootstrapping stability and convergence?
3. **Related Works:** Can you provide a more detailed discussion of the related literature regarding these specific weighting schemas?
4. **Alternative Weighting Schemes:** There is a rich literature on prioritized experience replay (PER) and weighted buffers. Have you experimented with other existing weighted buffer methods? If so, how did they perform compared to your specific Q-bound weighting?
5. **Prioritized Sampling vs. Trivial Samples:** Why does prioritizing samples that are expected to "converge faster" work well? Intuitively, this prioritizes "simple" samples. Does this not lead to a failure mode where the algorithm puts all of its weight on trivial transitions and ignores harder, yet critical, states?
6. **Comparison to Online RL Buffers:** Could you provide a more detailed comparison with weighted buffers from online RL (which often prioritize *high* error samples)? Are there previous works in offline RL that prioritize low-error/fast-
converging samples in this way?

**Limitations:**

Yes, the authors address this in the second paragraph of the conclusion.

**Strengths And Weaknesses:**

**1. Soundness**
* **Strengths:** The authors conduct detailed and extensive experiments on standard offline RL benchmarks, demonstrating empirical robustness.
* **Weaknesses:** The theoretical justification for the proposed weighting schemes is somewhat lacking, and the mathematical formulations for the weights feel heuristic-driven rather than derived from first principles. Additionally, while
the authors claim the two weighting schemas act independently, the theoretical support for this independence is weakly articulated.

**2. Presentation**
* **Strengths:** The overall motivation for analyzing loss landscape geometry in offline RL is clear.
* **Weaknesses:** A rigid, unified mathematical formulation of the complete Q-SAM framework is missing from the main text. The authors explain the two mechanisms (Weighted Sharpness and Weighted Samples) separately, but a cohesive explanat
ion of how they interact and are combined into a single final objective function is omitted. Furthermore, the notation leaves it ambiguous whether the sample weighting is applied to the sharpness loss as well, or exclusively to the Bellman error.

**3. Significance**
* **Strengths:** The experimental results are solid and indicate that the method addresses a highly relevant problem (extrapolation error and generalization in offline RL) effectively.
* **Weaknesses:** The significance is somewhat offset by the fact that the method introduces additional architectural complexity and hyperparameter tuning without strong fundamental guarantees.

**4. Originality**
* **Strengths:** The paper advances offline RL by bringing attention to an important optimization concept (SAM) and loss landscape geometry.
* **Weaknesses:** The core algorithmic novelty feels incremental, as it essentially combines two existing techniques (weighted sharpness and prioritized buffer/sampling). It feels that the paper adds another layer of complexity rather than introducing a fundamentally novel framework.

---

> ### Author Rebuttal · Authors · 2026-03-30
>
> We sincerely appreciate your effort in providing these valuable suggestions. We recognize our original description may have caused some misunderstanding, and hope the following clarifications will help resolve your concerns.
>
> **> 1. Mechanism Interaction (For Q1, Q2)**
>
> ---
> Algorithm: Q-SAM
>
> Input: Offline dataset $\mathcal{D}$, Q-network $Q_{\theta}$, error model $\Delta_{\varphi}$, policy $\pi_{\phi}$, parameters $\lambda, \rho, \alpha$.
>
> While not converged
>
> &ensp;&ensp;Sample batch $\mathcal{B}=\{(s_i,a_i)\}_{i=1}^{b} \sim \mathcal{D}$;
>
> 	Stage 1: Weighted Samples (for Critic)
>
> &ensp;&ensp;Compute $w_k(s,a) \propto \exp(-\gamma \Delta_{k-1}(s',\hat{a}')/\tau)$ using DisCor;
>
> &ensp;&ensp;**Update Critic: $\theta_{k+1}\leftarrow\arg\min_{\theta}\frac{1}{N}\sum_i^N w_k(s_i,a_i)(Q_\theta(s_i,a_i)-y_i)^2$;**
>
> 	Stage 2: Weighted Sharpness (for Actor)
>
> &ensp;&ensp;Compute perturbed policy gradient $g^{SAM}$ on the \textit{same} weighted batch $\mathcal{B}_w$;
>
> &ensp;&ensp;**Update Actor: $\phi_{t+1}=\phi_t-\alpha \left(\frac{\lambda}{1-\lambda}g^{SAM}_t + \frac{1-2\lambda}{1-\lambda}g^{Adam}_t\right)$;**
>
> Endwhile
>
> ---
>
> These two mechanisms constitute a "filter-then-optimize" cascading system:
> - **Weighted Samples** reweights the data distribution according to the cumulative Bellman error $\Delta_k$, identifying reliable samples where Q-values have converged (typically located at trajectory ends or leaf nodes).
> - **Weighted Sharpness** then seeks flat minima in the parameter space using these reliable samples.
>
> Both share the theoretical foundation of Q-value uncertainty quantification via $\Delta_k$, while operating orthogonally: one regularizes the data distribution for value learning, the other the parameter landscape for policy optimization.
>
> The algorithmic clarification and cascading mechanism will be detailed in Sec. 4.3 to resolve the noted ambiguity.
>
> **> 2. Prioritized Sampling vs. Trivial Samples (For Q4, Q5, Q6)**
>
> We may not have adequately conveyed the nuances of our sample weighting strategy, potentially creating the impression that we prioritize "simple" or "trivial" samples. We clarify that this is not the case:
>
> ① DisCor Prioritizes Reliable (Not Simple) Samples
>
> The DisCor uses the cumulative Bellman error bound:
>
> $$ \Delta_k(s,a) = |Q_k - \mathcal{T}Q_{k-1}| + \gamma \mathbb{E}[\Delta_{k-1}(s',a')] $$
>
> Samples are weighted by $w_k \propto \exp(-\Delta_k/\tau)$. Crucially:
> - **Not low TD-error**: $\Delta_k$ includes both current TD error *and* accumulated error. Even with high immediate TD error, **samples near trajectory ends (leaf nodes) receive high weights** because their future accumulation $\Delta_{k-1} \approx 0$.
> - **Not "easy" samples**: These are potentially high-reward, critical states (e.g., near-goal regions or stable periodic orbits in Hopper) that are difficult to reach but have converged Q-values due to shorter bootstrapping chains.
> - **Uncertainty-based**: The weighting reflects *estimation reliability*, not task difficulty.
>
> ② Why PER Fails in Offline RL
>
> Unlike online RL where high TD-errors indicate useful exploration, in offline RL they may signal OOD data that cannot be corrected through interaction. Prioritizing them (as PER does) thus causes overfitting to unreliable regions, exacerbating extrapolation error. We validate this by replacing our DisCor with PER while keeping the Weighted Sharpness component identical.
>
> ||ha-m|ha-mr|ha-me|ho-m|ho-mr|ho-me|wa-m|wa-mr|wa-me|Total|
> |:-|:-|:-|:-|:-|:-|:-|:-|:-|:-|:-|
> |CQL|47.0|45.0|95.6|59.1|95.1|99.3|80.8|73.1|109.5|704.5|
> |CQL+Q-SAM (use PER)|46.8±0.3|44.2±0.5|94.1±1.2|52.3±3.1|88.4±4.2|96.5±2.8|76.5±1.8|68.2±2.4|106.4±0.9|673.4|
> |CQL+Q-SAM (use DisCor)|47.3±0.2|45.6±0.2|95.6±0.5|70.3±2.5|95.5±5.6|109.8±1.7|84.2±1.2|84.4±4.2|112.3±0.3|745.0|
>
> Strikingly, these results reveal PER as counterproductive in offline RL: prioritizing high TD-error samples demonstrably degrades CQL performance below its vanilla baseline, exposing fundamental limitations of TD-error-based weighting under distribution shift. Conversely, DisCor demonstrates that prioritization of low cumulative errors consistently delivers substantial gains for effective sharpness-aware optimization.
>
> **> 3. Related Work (For Q3)**
>
> Experience replay in online RL splits into: (1) prioritization strategies—PER and ERO select samples by TD error, DisCor and ReMERT target corrective feedback and on-policiness, AER and LFIW emphasize frequently visited states; (2) sample diversity maintenance via Retrace, ReF-ER, RSDRB, and SEM.
>
> By contrast, offline RL rarely employs experience replay given the fixed dataset setting; instead, trajectory generation methods such as Synthetic Experience Replay, GTA, and PbRL are explored to expand coverage.
>
> We use DisCor not as a replay add-on but as a SAM-specific filter: suppressing high cumulative Bellman error transitions confines sharpness-aware optimization to converged regions, preventing noise from destabilizing flatness-seeking.

---

> > ### Author Rebuttal · Reviewer_aVSv · 2026-04-02
> >
> > I thank the authors for their detailed and thoughtful rebuttal. After reviewing the response, I find that my core concerns have been mostly addressed. Given these clarifications, I am happily maintaining my positive score for this paper.

---

> > > ### Author Response · Authors · 2026-04-06
> > >
> > > We sincerely thank you for your inspiring and valuable feedback! We are delighted to have resolved your concerns, and we will carefully refine and enhance the corresponding aspects in the revised manuscript!

---

### Official Review · Reviewer_uMJe · 2026-03-13

**Soundness:** 4
**Presentation:** 4
**Significance:** 4
**Originality:** 3
**Overall Recommendation:** 5
**Confidence:** 4

**Summary:**

Applying Sharpness-Aware Minimization (SAM) to off-line Reinforcement Learning (RL) remains an unexplored direction. This paper identifies a critical challenge when directly applying SAM to the bootstrapping mechanism employed in off-line RL. Therefore, this paper proposes Q-SAM to effectively adapt SAM for off-line RL. Specifically, Q-SAM introduces two novel and complementary mechanisms: Weighted Sharpness and Weighted Samples. A key advantage of Q-SAM is that it can be effectively integrated into various off-line RL methods. To validate Q-SAM, this paper conducts extensive experiments with different baseline methods on multiple datasets, accompanied by a parameter analysis.

**Compliance With Llm Reviewing Policy:**

Affirmed.

**Final Justification:**

My concerns have been addressed. Therefore, I maintain my original score.

**Key Questions For Authors:**

Please refer to “Weakness”.

**Limitations:**

yes

**Strengths And Weaknesses:**

Strengths:

Soundness:
This paper conducts comprehensive experiments with different baseline methods on multiple datasets, which validates the effectiveness and generalization ability of the proposed method. The proposed method involves a small number of hyperparameters, and a hyperparameter analysis is provided, which facilitates its adoption and further improvement.

Presentation:
This paper is well-written and has a clear logic. In particular, it provides a clear and understandable description of the motivation. This paper also includes proper citations of the related works.

Significance:
As an important technique for improving generalization, applying Sharpness-Aware Minimization (SAM) to off-line reinforcement learning (RL) remains an unexplored direction. The bootstrapping mechanism employed in off-line RL makes the application of SAM to off-line RL a non-trivial problem. This paper proposes an effective approach and identifies key directions for future improvement, which renders the work pioneering.

Originality:
To effectively adapt SAM for off-line RL, Q-SAM introduces two complementary components: Weighted Sharpness and Weighted Samples. These two components correspond to the Actor and the Critic in RL methods, respectively. Experimental results demonstrate the superiority of Q-SAM over SAM and the effectiveness of each proposed component. This paper possesses reasonable originality in both the identification of the problem and the formulation of its solution.

Weaknesses:

1. A clearer explanation of Eq. (6) can be provided.
2. A brief description of how the two mechanisms, Weighted Sharpness and Weighted Samples, work in conjunction (as shown in Appendix Section A-Algorithm 4) can be included in Section 4.3 of the main text.
3. The design of Weighted Sharpness is inspired by Weighted Sharpness-Aware Minimization, and a version of Weighted Sharpness that is more specifically tailored for off-line RL could be proposed in the future. However, it is worth noting that a single paper cannot address all problems.

---

> ### Author Rebuttal · Authors · 2026-03-30
>
> We sincerely thank you for your encouraging and highly professional evaluation. Your insightful comments have been instrumental in improving the clarity and completeness of our work. Below are our detailed point-by-point responses to your questions:
>
> **> 1. Clarification on Equation (6)**
>
> Equation (6) computes the cumulative Bellman error upper bound $\Delta_k$ through temporal composition:
>
> $$\Delta\_k = \underbrace{|Q\_k - \mathcal{T}Q\_{k-1}|}\_{\text{current residual}} + \underbrace{\gamma P^{\pi\_{k-1}}\Delta\_{k-1}}\_{\text{discounted future error}}$$
>
> The first term captures the immediate Bellman residual at the current state-action pair, while the second term propagates the error from subsequent states via the backup operator $P^{\pi_{k-1}}$, discounted by $\gamma$. This recursive structure aggregates uncertainty backward through trajectories without requiring knowledge of $Q^*$.
>
> In implementation, Equation (6) translates directly to the following computation (see `calc_target_errors` in the supplementary code):
> ```python
> target_errs1 = (curr_qs1 - target_qs).abs() + (1.0 - dones) * self._gamma * next_errs1
> ```
> Here, `(curr_qs1 - target_qs).abs()` computes the current residual, and `next_errs1` (obtained from the target error network) provides $\Delta_{k-1}$. The term `(1.0 - dones)` masks terminal states where bootstrapping should terminate.
>
> To derive the sample weights $w_k$ from this error bound, we first estimate the next-step error via the target network:
> ```python
> next_errs1, next_errs2 = self._target_error_net(next_states, next_actions)
> ```
> then compute self-normalized importance weights through exponential transformation and Softmax:
> ```python
> x1 = -(1.0 - dones) * self._gamma * next_errs1 / (self._tau1 * self.tau_scale)
> imp_ws1 = F.softmax(x1, dim=0)  # This yields w_k
> ```
> This process will be detailed in Section 4.2 of the revised manuscript to clarify how Equation (6) and Equation (7) bridges theoretical error bounds with practical sample weighting.
>
> **> 2. Clarification on Algorithmic Integration of Both Mechanisms in Section 4.3**
>
> Thank you for the suggestion. We will include the following algorithmic description in Section 4.3 to clarify how the two mechanisms operate in conjunction:
>
> ---
> Algorithm: Q-SAM
>
> Input: Offline dataset $\mathcal{D}$, Q-network $Q_{\theta}$, error model $\Delta_{\varphi}$, policy $\pi_{\phi}$, parameters $\lambda, \rho, \alpha$.
>
> While not converged
>
> &ensp;&ensp;Sample batch $\mathcal{B}=\{(s_i,a_i)\}_{i=1}^{b} \sim \mathcal{D}$;
>
> 	Stage 1: Weighted Samples (for Critic)
>
> &ensp;&ensp;Compute $w_k(s,a) \propto \exp(-\gamma \Delta_{k-1}(s',\hat{a}')/\tau)$ using DisCor;
>
> &ensp;&ensp;**Update Critic: $\theta_{k+1}\leftarrow\arg\min_{\theta}\frac{1}{N}\sum_i^N w_k(s_i,a_i)(Q_\theta(s_i,a_i)-y_i)^2$;**
>
> 	Stage 2: Weighted Sharpness (for Actor)
>
> &ensp;&ensp;Compute perturbed policy gradient $g^{SAM}$ on the \textit{same} weighted batch $\mathcal{B}_w$;
>
> &ensp;&ensp;**Update Actor: $\phi_{t+1}=\phi_t-\alpha \left(\frac{\lambda}{1-\lambda}g^{SAM}_t + \frac{1-2\lambda}{1-\lambda}g^{Adam}_t\right)$;**
>
> Endwhile
>
> ---
>
> This procedure reveals the **"filter-then-optimize"** dependency: Weighted Samples first reweights the batch using cumulative Bellman error $\Delta_k$ (identifying converged trajectory segments), and Weighted Sharpness subsequently performs flatness-seeking exclusively on this stabilized data distribution. While both leverage the same Q-value uncertainty bound $\Delta_k$, they operate orthogonally—one regularizing the data distribution for value learning, the other the parameter landscape for policy optimization.
>
> **> 3. Future directions for Weighted Sharpness**
>
> Indeed, our Weighted Sharpness builds upon WSAM. While the current $\lambda$-based weighting effectively balances flatness and convergence, we agree that offline-RL-specific adaptations—such as **dynamic $\lambda$ scheduling** based on value convergence rates or trajectory-level uncertainty—represent promising future work.
>
> Furthermore, **sample-adaptive sharpness weighting**—where the coefficient $\lambda$ is modulated **per-sample** based on local gradient variance or epistemic uncertainty estimates—could enable finer-grained control over the flatness-convergence trade-off across heterogeneous regions of the loss landscape.
>
> We will include this discussion in Section 6 of the revised manuscript.

---

> > ### Author Rebuttal · Reviewer_uMJe · 2026-04-03
> >
> > I thank the authors for their clarification. My concerns have been addressed. The answer to Question 1 explains the Eq.(6). The answer to Question 2 explains how the two mechanisms work together. The answer to Question 3 explains the future directions for improving the Weighted Sharpness.

---

> > > ### Author Response · Authors · 2026-04-06
> > >
> > > We sincerely thank you for your thoughtful and valuable feedback! We are glad to have addressed your questions, and we will carefully implement and strengthen these improvements in the final version!

---

### Official Review · Reviewer_u4yj · 2026-03-16

**Soundness:** 3
**Presentation:** 3
**Significance:** 3
**Originality:** 3
**Overall Recommendation:** 4
**Confidence:** 3

**Summary:**

This paper revisits the optimization landscape of offline RL and identifies a fundamental conflict between generic sharpness minimization and the bootstrapping mechanism. To harmonize these objectives, the authors apply Sharpness-Aware Minimization (SAM), which has recently shown strong generalization benefits in supervised learning to offline RL. They proposed a framework, namely Q-SAM, that treats sharpness as a weighted objective and selectively prioritizes samples that are most suitable for sharpness-aware optimization based on Q bounds. Experiments demonstrate that Q-SAM achieves superior generalization and stability across diverse benchmarks.

**Compliance With Llm Reviewing Policy:**

Affirmed.

**Final Justification:**

My concerns have been addressed. I maintain my score, leaning toward accepting.

**Key Questions For Authors:**

I have several questions:

- When combining SAM with Weighted Samples, how to compute $w_k$ in real-world applications? I also did not see how to choose the hyperparameter $\lambda$?

- It remains unclear why the individual components of Q-SAM appear to degrade performance when isolated, yet yield significant gains when combined. Could the authors elaborate on the synergistic effect between these two components?

- Could the authors explain better why exploring SAM optimization for the value function is an interesting direction for future research?

**Limitations:**

Yes

**Strengths And Weaknesses:**

This paper is well written and well structured. It addresses the generalization, an important and challenging problem in offline RL when facing the distribution shift. Departing from traditional methods, this paper explores offline RL through the lens of optimization geometry, investigating if SAM’s flatness-seeking properties can enhance generalization. By identifying geometric control as a critical frontier in the field, the authors make a significant contribution.

They introduce Q-SAM, a framework that resolves inherent objective conflicts by dynamically weighting sharpness based on Q-value reliability. Their experiments confirm that Q-SAM improves generalization and stability across diverse benchmarks.

---

> ### Author Rebuttal · Authors · 2026-03-30
>
> We sincerely thank you for your thorough evaluation and constructive feedback, which has helped us significantly clarify the technical details and strengthen the presentation. Below are our point-by-point responses.
>
> **> 1. Computation of $w_k$ and Selection of $\lambda$**
>
> ① In practice, the computation of $w_k$ follows a **five-step process**:
>
> **Step 1: Estimate the next-step cumulative error $\Delta_{k-1}$**
>
> Obtain the cumulative Bellman error for the next state-action pair from the target error network:
>
> 	next_errs1, next_errs2 = self._target_error_net(next_states, next_actions)
>
> **Step 2: Compute exponential weights**
>
> Calculate $-\gamma \cdot \Delta_{k-1} / \tau$, where the term (1.0−dones)  masks out terminal states to prevent error propagation:
>
> 	x1 = -(1.0 - dones) * self._gamma * next_errs1 / (self._tau1 * self.tau_scale)
>
> **Step 3: Self-normalization to obtain $w_k$**
>
> Apply Softmax across the batch to ensure the weights sum to 1:
>
> 	imp_ws1 = F.softmax(x1, dim=0)
>
> **Step 4: Compute cumulative error targets (for training the error network)**
>
> Combine the current TD error with the discounted next-step error to compute the target for updating $\Delta_k$
>
> 	target_errs1 = (curr_qs1 - target_qs).abs() + (1.0 - dones) * self._gamma * next_errs1
>
> **Step 5: Apply $w_k$ to the Critic Loss**
>
> Reweight the MSE loss using the computed importance weights:
>
> 	q1_loss = torch.mean((current_q1 - target_q).pow(2) * weights1)
>
> The above process can be verified in the source code provided in the supplementary materials, and we will incorporate this description into the revised manuscript to improve clarity.
>
> ② $\lambda \in [0,1)$ controls the trade-off between flatness and loss minimization:
> - $\lambda=0.5$ corresponds to standard SAM; $\lambda<0.5$ prioritizes loss minimization (more aggressive), while $\lambda>0.5$ emphasizes flatness.
> - Empirical guidance: Tab. 10 (Appendix B.5) shows that $\lambda=0.1$ generally yields the best results across 7/9 tasks for CQL and IQL baselines, suggesting that **conservative sharpness weighting is preferable in offline RL to avoid exacerbating extrapolation errors**.
> - However, $\lambda=0.9$ occasionally helps escape local optima in specific tasks (e.g., hopper-mr), indicating that the optimal $\lambda$ balances between reducing extrapolation errors and avoiding local optima.
>
> **> 2. Synergistic Effect Between Components**
>
> ① Why isolation degrades performance
>
> - **Weighted Sharpness alone** mitigates the tension with convergence but still applies SAM uniformly to all samples, including high-error transitions. Without error-based filtering, the optimizer seeks flat regions around unreliable Q-estimates (particularly early-trajectory states), "flattening" noise rather than signal—explaining the -14.2 drop when removing Weighted Samples (Tab. 4).
> - **Weighted Samples alone** filters reliable data via $\Delta_k$, but leaves the sharpness objective uncalibrated. This forces the policy to pursue flat minima while Q-values are still evolving, creating a mismatch between value convergence and geometric constraints—causing the -8.5 degradation.
>
> ② Synergistic mechanism
>
> Together, they form a "filter-then-optimize" cascade:
> - **Weighted Samples as preconditioning**: By filtering transitions through cumulative Bellman error $\Delta_k$, it creates a clean optimization landscape where samples exhibit converged Q-values. This removes the non-stationarity that would otherwise poison sharpness estimation.
> - **Weighted Sharpness as calibrated regularization**: Operating on this subset, flatness-seeking induces policy smoothness without destabilizing the optimization process.
>
> Sharpness estimation requires computation over converged Q-values, while preventing overfitting to the filtered subset requires smoothness regularization. Neither function suffices alone because bootstrapping noise must be filtered before geometric constraints apply.
>
> **> 3. SAM for Value Function**
>
> **Untapped theoretical potential**: Extending SAM to value functions promises dual regularization: flat Q-minima could stabilize value estimates against OOD queries at the source (tackling over-optimism directly), complementing policy smoothing which only addresses the symptom (erratic actions).
>
> **Synergistic extension**: Optimizing both policy and value landscapes would provide complementary robustness benefits. While policy SAM ensures smooth action selection, value SAM independently stabilizes Q-estimates under distribution shift, together creating a more resilient optimization landscape.
>
> **Feasible solutions under exploration**: We are investigating Truncated Bellman Backup (n-step=3 to limit error propagation) and Target Network SAM (applying sharpness only to slowly-updated target networks). These adaptations would enable safe flatness-seeking in pure critic settings, validating that Q-SAM's core principle—stabilizing data distribution while optimizing value geometry—extends naturally to actor-free TD learning.

---

> > ### Author Rebuttal · Reviewer_u4yj · 2026-04-05
> >
> > I thank the authors for their responses to my concerns. My concerns have now been addressed. I keep my score.

---

> > > ### Author Response · Authors · 2026-04-06
> > >
> > > We sincerely thank you for your thorough and constructive feedback! We are delighted to have addressed your concerns, and we will carefully enhance these aspects in the revised manuscript!

---

### Decision · Program_Chairs · 2026-04-30

**Decision:**

Accept (regular)

**Comment:**

The paper proposes Q-SAM, an optimizer design for improving generalization in offline RL by adapting the sharpness-aware minimization (SAM) approach to the bootstrapped setting. SAM is one of the most influential work in turning generalization analysis into practical optimization objective to improve test-time performance. Reviewers found the problem important and the optimization perspective novel and meaningful, especially given that SAM has been little explored in offline RL (and RL in general). The empirical study was generally viewed as strong. The main concerns focused on clarity of formulation and credit assignment among components, but reviewers indicated that their concerns were resolved after rebuttal. The reviews are uniformly positive after discussion, and I therefore support acceptance.